# The multi-dimensional challenges of controlling respiratory virus transmission in indoor spaces: Insights from the linkage of a microscopic pedestrian simulation and SARS-CoV-2 transmission model

**Büsra Atamer Balkan**[1☺], **You Chang**[1☺], **Martijn Sparnaaij**[2], **Berend Wouda**[3], **Doris Boschma**[3], **Yangfan Liu**[1¤], **Yufei Yuan**[2], **Winnie Daamen**[2], **Mart C. M. de Jong**[1], **Colin Teberg**[4], **Kevin Schachtschneider**[4], **Reina S. Sikkema**[5], **Linda van Veen**[3], **Dorine Duives**[2‡*], **Quirine A. ten Bosch**[1‡*]

**1** Quantitative Veterinary Epidemiology, Wageningen University & Research, Wageningen, The Netherlands, **2** Department of Transport & Planning, Delft University of Technology, Delft, The Netherlands, **3** Gamelab, Delft University of Technology, Delft, The Netherlands, **4** Steady State Scientific Computing, Chicago, Illinois, United States of America, **5** ViroScience, Erasmus Medical Center, Rotterdam, The Netherlands

☺ These authors contributed equally to this work.
¤ Current address: Department of Veterinary and Animal Sciences, University of Copenhagen, Copenhagen, Denmark
‡ These authors are joint senior authors on this work.
* d.c.duives@tudelft.nl (DD); quirine.tenbosch@wur.nl (QtB)

## Abstract

SARS-CoV-2 transmission in indoor spaces, where most infection events occur, depends on the types and duration of human interactions, among others. Understanding how these human behaviours interface with virus characteristics to drive pathogen transmission and dictate the outcomes of non-pharmaceutical interventions is important for the informed and safe use of indoor spaces. To better understand these complex interactions, we developed the Pedestrian Dynamics—Virus Spread model (PeDViS), an individual-based model that combines pedestrian behaviour models with virus spread models incorporating direct and indirect transmission routes. We explored the relationships between virus exposure and the duration, distance, respiratory behaviour, and environment in which interactions between infected and uninfected individuals took place and compared this to benchmark 'at risk' inter-actions (1.5 metres for 15 minutes). When considering aerosol transmission, individuals adhering to distancing measures may be at risk due to the buildup of airborne virus in the environment when infected individuals spend prolonged time indoors. In our restaurant case, guests seated at tables near infected individuals were at limited risk of infection but could, particularly in poorly ventilated places, experience risks that surpass that of bench-mark interactions. Combining interventions that target different transmission routes can aid in accumulating impact, for instance by combining ventilation with face masks. The impact of such combined interventions depends on the relative importance of transmission routes, which is hard to disentangle and highly context dependent. This uncertainty should be considered when assessing transmission risks upon different types of human interactions in

**Data Availability Statement:** The development of PeDViS is part of a research project that develops decision support tools for practitioners to limit SARS-CoV-2 transmission inside their venues. An open-access web-based simulation environment was created, named the SamenSlimOpen App (SSO app: https://www.samenslimopen.nl/de-tool/ ). The PeDViS model is at the core of this app (Section C in S1 Text). All code for the PeDViS model and data to recreate the described experiments are openly available on Gitlab (https://git.wur.nl/sso-public/pedvis).

**Funding:** This publication is part of the project SamenSlimOpen (10430022010018, awarded to QAtB) of the research programme COVID-19 Programma, which is financed by the Dutch Research Council (NWO) and ZonMw. BAB, YC, MS, BW, DB, YY, WD, MdJ, LvV, DD, and QAtB received salary from this grant. CT and KS were hired as software developers and paid for by the grant. The funders did play no role in the study design, data collection and analysis, decision to publish, or the preparation of the manuscript.

**Competing interests:** The authors have declared that no competing interests exist.

indoor spaces. We illustrated the multi-dimensionality of indoor SARS-CoV-2 transmission that emerges from the interplay of human behaviour and the spread of respiratory viruses. A modelling strategy that incorporates this in risk assessments can help inform policy makers and citizens on the safe use of indoor spaces with varying inter-human interactions.

## Author summary

With most infections happening indoors, indoor spaces played an important role in the spread and control of SARS-CoV-2. Indoor transmission and the impact of interventions targeted at these spaces are hard to predict due to the interplay of diverse inter-human interactions, host factors, virus characteristics, and the local environment. Mathematical models can help disentangle such complex processes. Here, we introduce a model that simulates viral spread in indoor spaces by combining models on detailed human movements and interactions with models that simulate the spread and uptake of viruses through direct and indirect transmission routes. We use a restaurant setting as a case-study and illustrate that, while common distancing measures hold for infection prevention during relatively short interactions, transmission may occur over longer distances if infected individuals spend more time in a space, particularly if poorly ventilated. The effects of intervention measures are tightly coupled to the transmission route they target and the relative importance of this route in a specific scenario. Uncertainty around the latter should be considered when assessing transmission risks. The model can be adapted to different settings, interventions, levels of population immune protection, and to other virus variants and respiratory pathogens. It can help guide decision making on effective mitigation of virus transmission in indoor spaces.

## 1. Introduction

With transmission estimated to be 18 times more likely to happen indoors than outdoors [1], indoor spaces played a focal role in the control of SARS-CoV-2 transmission [2–8]. This risk of transmission can however vary greatly across settings, depending on the context, indoor environment, variation in individual behaviours, infectiousness and susceptibility (e.g., due to immune protection) to the virus, as well as the level of adherence to intervention measures. **Understanding how the interplay of human behaviour and viral spread in different environments affects SARS-CoV-2 transmission in indoor spaces is important for the design of effective mitigation strategies.**

The efficiency of indoor transmission of respiratory viruses depends on several factors that interact in non-straightforward ways. In general, the likelihood and extent of secondary infections that result from a single introduction depend on three things: the contact structure, individual host and virus characteristics, and the environment in which the contacts take place. First, transmission is driven by the duration, closeness, and number of contacts the infectious individual has while visiting the indoor space [9,10]. Crowd monitoring tools have been used during the COVID-19 pandemic to record the frequency and duration of contacts and inform the topology of human interactions in different settings. These studies show, amongst other things, that the changes in the interaction patterns as a result of COVID-19 pandemic are very context dependent [11,12], and that habitual interaction patterns are difficult to change [13]. Second, how likely each of these contacts is to result in infection depends on the characteristics

of the infected individual (infectiousness, respiratory behaviours), the susceptibility of the contact individuals (as a result of immunity and other individual characteristics), and how effectively the virus spreads from one individual to the next. For respiratory viruses, transmission can generally happen through i) droplet spread (large viral-laden droplets that fall to the ground rapidly), ii) aerosol spread (small viral-laden droplets that have the potential to remain airborne for some duration of time), and iii) fomite transmission (i.e., when contaminated surfaces act as intermediary vectors that result in virus exposure when individuals touch them). How effective each of these routes is, depends on the virus, the indoor conditions (i.e., temperature, ventilation, and humidity which may affect the persistence of viruses in their environments), and the closeness, frequency, and duration of contacts.

Many non-pharmaceutical interventions (NPIs) are targeted at public indoor spaces, including physical distancing, the use of face masks, hygiene measures, improved ventilation, and limiting crowding [2–5]. Although we know these NPIs to be effective in some spaces [14–19], predicting their impact in various settings and epidemiological contexts is not straightforward. In part, this is because NPIs differ by the main transmission route that they interfere with. While face masks mostly prevent droplet spread, improved ventilation predominantly interferes with the concentration of virus-laden aerosols. The impact of NPIs therefore depends on the context-specific relative contributions of different transmission routes. This also affects predictions on composite effects of NPIs, which is likely to be highest if combinations of NPIs are sought that affect complementary transmission routes, depending on the virus. Lastly, the level of compliance to the different NPIs may greatly determine their impact. It is this complex interaction between context-specific drivers of transmission, the choice and nature of NPIs, and the level of compliance thereof that make it challenging to preempt the success of intervention strategies.

Mathematical models can help decipher these complex interactions. One can gain understanding on the, often non-linear, relationships that drive the spread of pathogens by combining mechanistic understanding of the transmission process on a population-level with knowledge on the distinct parts of the transmission process (typically on the individual or pathogen-level). Most mathematical models developed during the COVID-19 pandemic evaluate interventions at national or subnational levels [20–24]. Other efforts focus on smaller scale transmission, such as hospitals [25]; supermarkets [26]; educational settings [27,28] and work environments [29,30]. Due to the central role they play in transmission and the fact that most control strategies are targeted at these settings, indoor spaces have started to receive more attention from modellers for SARS-CoV-2 and other respiratory pathogens [31–35].

A particular goal of such indoor transmission models is to better understand how the heterogeneity of encounters in indoor spaces affect transmission and influence the effectiveness of NPIs [36–39]. In one group of airborne transmission models, the Wells-Riley models, one assumes that infectious particles are well mixed in the indoor space. As a consequence, the amount of virus that individuals are exposed to is independent of their distance to the infectious individual and solely depends on the duration of this contact. Using these models, the effect of e.g., restricting occupancy and total event duration can be assessed [40]. Expansions of the Wells-Riley model have been proposed that allow for individual heterogeneity in infectiousness and respiratory activities [37,39,40], for spatial variation of the virus distribution in the environment [41,42], and the inclusion of multiple transmission routes [27,36]. The multi-route transmission models consider the transmission also via droplets and fomites and shed light on how the relative importance of transmission routes depends on the duration and distance of infectious contacts [30,33,36]. A final class of indoor transmission models follow the computational fluid dynamics (CFD) principles and simulates the flow of particles in time and space [28,43,44]. (See Section A in S1 Text for an overview of indoor transmission models).

Most indoor transmission models described above assume simple, static interactions between individuals. Some recent advances have been made to incorporate the dynamic nature of human interactions and explore its impact on transmission [45,46]. The first of these models use descriptions of human behaviour such as contact duration [46,47] and couple these with simple rules on transmission risks, such as assuming a linear relationship between exposure duration and infection risk [26,46–50]. These parsimonious descriptions of pedestrian movement are helpful to build general understanding of transmission potential in crowded spaces, but are less useful to disentangle the impact of interventions that affect the contact structures (e.g., routing, distancing, cohorting) and associated transmission risks. Individual-based models allow for the simulation of more realistic, diverse, context-based movements by including activity spaces (i.e., where and when do we spend our time) and pathfinding (i.e., how to reach a destination without colliding into objects or other individuals). Such models, when carefully calibrated to empirical observations, can contribute to our understanding of the relationships between human-building interactions and their potential impact on virus spread and exposure [51,52]. (See Section A in S1 Text for an overview of pedestrian models and their applications in infectious disease epidemiology).

While great advancements have been made in the modelling of indoor respiratory virus transmission, challenges remain in linking simulated virus exposure to epidemiologically meaningful infection risks and doing so across the range of possible settings and human interactions. Models that combine context-based human activity (as determined by activity patterns and route choice of individuals present in an indoor space) with detailed SARS-CoV-2 spread, viral exposure, and consequent infection risks and enumerate the levels of uncertainty surrounding these outcomes, may form a valuable addition to the existing model ecosystem and help further guide recommendations on the safe use of public spaces.

The objective of this paper is to examine how behavioural, viral, and the indoor environmental factors interplay in determining SARS-CoV-2 transmission risks and the relative impact of non-pharmaceutical interventions in indoor environments.

To do so, we developed a combined Pedestrian Dynamics—Virus Spread model (PeDViS model) that combines an established pedestrian movement model and a multi-route spatially explicit viral transmission model. Recent insights from pedestrian modelling, virology, epidemiology, and IT-design are combined to develop this open-access software package to model the transmission of SARS-CoV-2 in indoor spaces. In particular, an expert-driven activity assignment model [53] is coupled with a force-based microscopic simulation model (NOMAD) and a virus spread model (Model for Quantifying Viruses in Environments, QVEmod). Here, using a restaurant as a case study, we investigate how human interactions propagate transmission risks in indoor spaces and illustrate how these estimates are affected by differences in contact structures, the indoor environment, and the interventions in place. We highlight the importance of the efficiency of different transmission routes by illustrating how uncertainty surrounding their relative contributions affects our ability to model transmission risks and predict the impact of (the combined application of) NPIs.

## 2. Model overview

In this research, we designed and implemented a combined model coined PeDViS. PeDViS chains an expert-driven strategic choice model with an existing microscopic pedestrian simulation model (NOMAD) [54,55] and an epidemiological model for Quantifying Viruses in Environments (QVEmod), see Fig 1.

The first model in the modelling chain transforms user input regarding the context, spatial layout, population, and demand into a set of personalised activity schedules [53]. The strategic

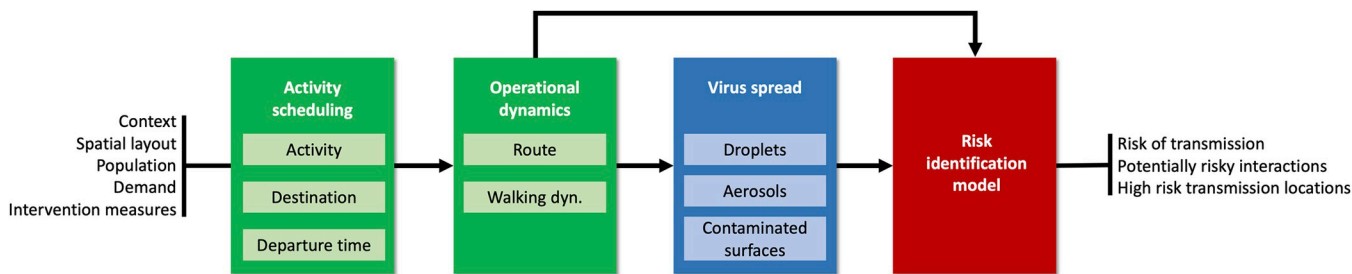

**Fig 1. Model chain to simulate SARS-CoV-2 transmission in indoor spaces.**

choice model consists of multiple sub-models, which jointly determine the activity choices, destination choices, and departure time choices of each pedestrian in the simulation model. To derive the personal schedules, the strategic choice model also assigns personal characteristics to each agent. Here, user-specified settings that impact activity choices are taken into account: for example, in a restaurant environment, the model considers the availability of toilets and paying at the table.

The second model (NOMAD) uses activity schedules and personalised characteristics to determine the operational movement behaviour of each individual. The operational movement behaviour features two sub-models, which determine the (best) route for each activity in a pedestrian's activity schedule towards each destination and their corresponding walking dynamics (i.e. walking velocity and acceleration) along the route. Both route and operational walking dynamics models take user-specified measures to limit SARS-CoV-2 transmission into account: for instance, following the physical distancing rules has an impact on collision avoidance behaviour, which eventually impacts the operational walking dynamics. The result of the second model is detailed dynamic trajectory information for each individual within the space.

The third model (QVEmod) uses these trajectories, combined with epidemiological attributes of the individuals (most notably the infectious status of individuals and respiratory behaviour), to simulate the spread of the virus in the environment and the extent to which susceptible individuals concurrently (or shortly after) present in the same space get exposed to it. The infectious status of individuals can be randomly assigned or targeted towards specific agents depending on the design of the simulation experiment. How SARS-CoV-2 is distributed over time and in space is modelled as the accumulation of the virus in the environment, both within the airborne particles (i.e., droplets and aerosols) in the air and on contaminated surfaces (fomites). This is informed by empirical data on the emission of the virus, the stability of both the virus and the airborne particles that carry it, and the uptake (i.e., through inhalation or by touching fomites) by individuals of virus through air and fomites. Susceptible individuals may get exposed to the virus by inhaling airborne particles or touching contaminated surfaces. This modelling step results in estimates of relative virus contamination at any location in the indoor space at each moment in time as well as individuals' exposure to virus via each of three considered transmission routes: droplets, aerosols, and fomites.

The final model, Risk Identification Model, assesses each individual's risk of becoming infected with SARS-CoV-2 based on the total amount of virus they are exposed to by applying dose-response relationships. After calculating the infection risk at the individual level, we use Monte Carlo simulation to estimate the number of newly infected individuals.

The details of model structure and equations are provided in the Methods section, and the details on model parametrization are presented in Section A in S1 Text.

## 3 Results

### 3.1 Virus spread between static contacts

The model assesses individuals' exposure over time and distinguishes the relative contribution of transmission routes to overall exposure in different settings as they arise from human interactions. To disentangle the interplay between the several factors that affect virus transmission, we first illustrate the working of QVEmod for various static contacts. We conduct simulation experiments (the term *experiments* used throughout the text refers to *computer simulation experiments*) to examine the three main factors of QVEmod namely the impact of i) the intensity of a contact (section 3.1.1), ii) respiratory activities (section 3.1.2), and iii) interventions implemented in the PeDViS model (section 3.1.3). The relationship between exposure and infection risk is likely to be non-linear (typically S-shaped) and different between routes (see details in 5.3.5). Relative differences in exposure should therefore not be interpreted as proportional to differences in infection risks.

In the following static contact experiments, the results are presented relative to a benchmark contact. The benchmark contact is defined as a scenario where susceptible and infectious individuals arrive concurrently in an indoor space and have a contact at a distance of 1.5 metres for 15 minutes, which is broadly used as an indicator of 'a risky contact' [4]. In that case, both infectious and susceptible individuals are assumed to talk and breathe both for 50% of the time each (akin to an interaction in a restaurant for instance), and the indoor space has an average ventilation rate of 3 air changes per hour (ACH). In section 3.1.1, we examined the impact of three determinants of contact intensity on exposure: duration, distance, and the time an infectious individual spent in the space prior to the contact. Then, in section 3.1.2, we examined the impact of different respiratory activities, namely breathing, talking and singing on relative emission and exposure. Lastly, in section 3.1.3, we examined the impact of different ventilation levels and face masks on exposure in a benchmark contact. The details of experiment settings are provided in Section B in S1 Text.

**3.1.1 The impact of contact intensity on exposure.** First, we examine the impact of contact intensity on virus exposure resulting from a static contact. We examine three determinants of contact intensity: duration, distance, and the time an infectious individual spent in the space prior to the contact. We distinguish the exposure to three routes, where droplet transmission is considered a direct route, and aerosols and fomites are considered indirect routes as the buildup of virus in the environment via these routes can potentially contribute to exposure after the infectious individual has left. Contacts at shorter distance than 1.5m result in substantially larger exposures with a 3-fold increase at 1 metre (13-fold at 0.5 metre) (Fig 2A). Exposure at longer distances diminishes quickly, with exposure at 2 metres being 3-fold lower than the benchmark of 1.5m. At 1.5m distance, 78% of exposure is expected to be attributable to aerosolized virus (here defined as those particles smaller than 10 um) (Fig 2A). Exposure at short distance (0.5m) is dominated by droplet transmission routes, although short range aerosolized viruses may also contribute meaningfully to overall exposure. Prolonged contacts are associated with an increase in exposure. A static contact at 1.5 metres for 1 hour is expected to result in exposure 9-fold higher relative to a 15 minute contact (Fig 2B). The contribution of indirect transmission routes increases with contact duration, highlighting the impact of virus buildup in environments. In other words, the impact of contact duration on exposure is larger than what would be expected if only direct routes played a role in transmission. A similar effect is seen when the infectious individual has spent 3 hours in the space preceding the contact. In such a scenario, exposure following a benchmark contact would be 2.5-fold higher than in our default scenario (Fig 2A and 2C). This increase is driven by a buildup of viruses (aerosolized) in the environment. These particles make up 88% of the expected exposure versus 78% under the baseline scenario. As individuals in this experiment stand still and do not share any

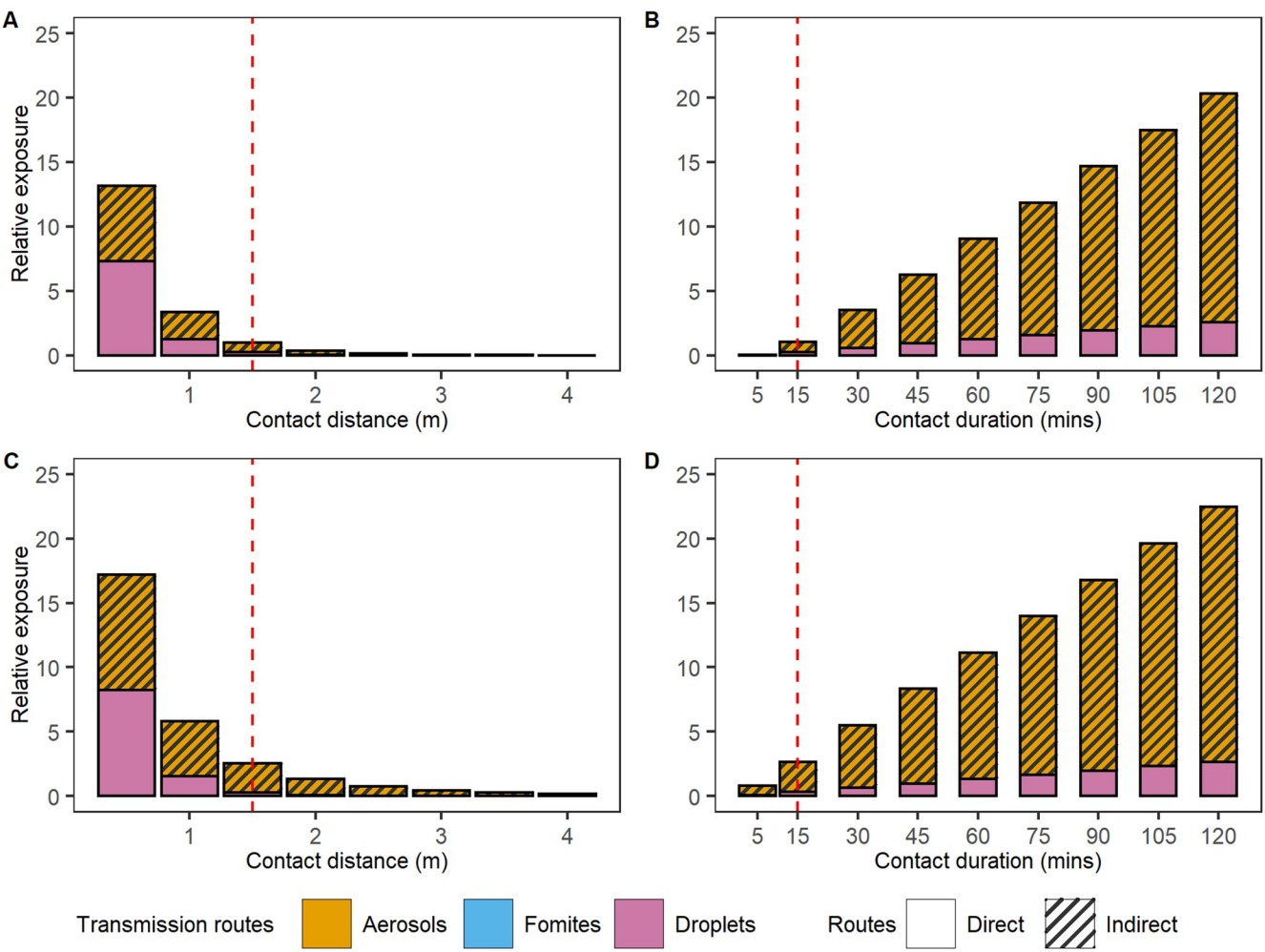

**Fig 2. Effect of contact intensity on exposure and the relative contribution to exposure of transmission routes.** A) exposure for 15 minutes at increasing distance, B) exposure at 1.5 metres for an increasing duration, C and D) as A and B but when the infectious individual was present 3 hours prior to the contact occurring, allowing for a buildup of virus in the environment. Red dashed lines show the contact with 1.5 metres and 15 mins. Exposure is shown relative to this benchmark, in a scenario of concurrent arrival of the infectious and susceptible individuals (as shown in A and B). For instance, a relative exposure of 25 means that overall exposure is 25 times that of the exposure of a benchmark contact. Individuals do not share common surfaces in this experiment, thus exposure from the fomites routes is negligible.

common surfaces, the exposure from the fomites routes is negligible in Fig 2. These first analyses with QVEmod illustrate that RIVM (Dutch National Institute for Health and Environment) guidelines regarding risky contacts, i.e., 1.5 metre distance and less than 15 minutes of exposure, provide good guidance to minimise exposure risks, provided infectious individuals have not convened in the same space for an extended period of time.

**3.1.2 The impact of respiratory activities on exposure.** The emission of virions per hour from talking and singing is assumed to be respectively 14 times and 16 times higher than from breathing [56] (Fig 3A). The make-up of the emitted particles (i.e., proportion aerosols and droplets) also varies depending on the respiratory behaviour, and are estimated to be 17% aerosols upon talking and 7% upon singing, relative to 98% upon breathing (Section E in S1 Text). Considering these factors in QVEmod, virus exposure upon 15 minutes of talking and singing was estimated to be about four times and eight times higher than upon breathing, respectively (Fig 3B). Notably, the contribution of aerosols to the overall exposure is estimated

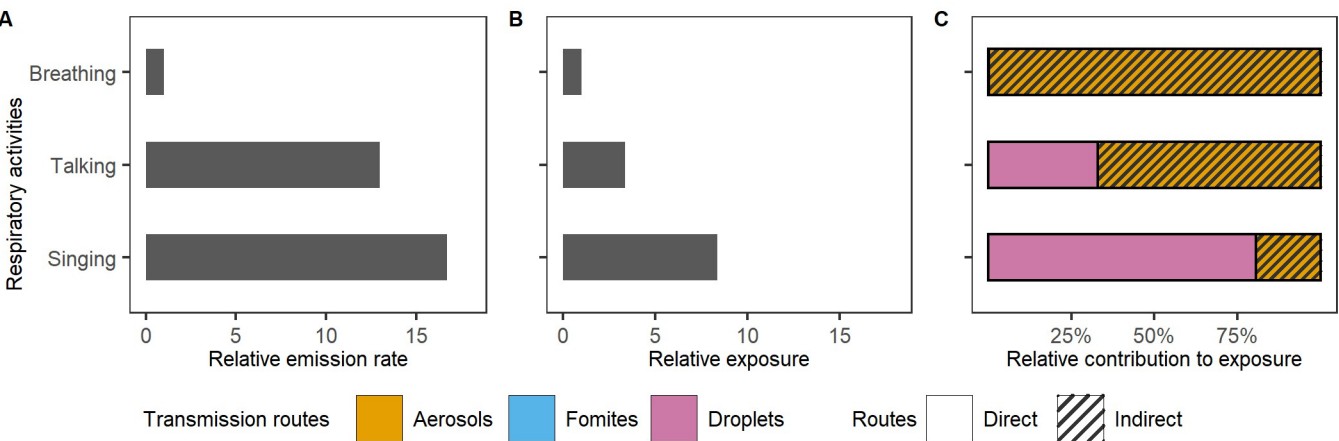

**Fig 3. Effect of different respiratory activities on exposure.** A) The relative emission rate of virions and B) the relative exposure during talking and singing continuously for 15 mins, relative to breathing. C) The relative contribution of the three transmission routes to the individual's exposure while breathing, talking and singing. Both the infectious and susceptible individuals are assumed to perform the respective respiratory activity. Individuals do not share common surfaces in this experiment, thus exposure from the fomites routes is negligible.

to be lower upon active respiratory activities (Fig 3C). However, due to different measuring methods and equipment, the quantity and partition of aerosols and droplets generated during different respiratory activities are inconsistent among studies [57,58] and may well differ between individuals of different age and gender [59,60].

**3.1.3. The impact of interventions on exposure.** Beyond distancing measures, improved ventilation and wearing face masks are common interventions in indoor spaces. Here, we examined the impact of both intervention measures across a range of possible efficacies by simulating the impact of these two interventions on exposure upon a static benchmark contact. With an ACH as high as 24, ventilation can result in a maximum reduction of overall exposure of about 65% in this static example (Fig 4A). Increasing the ACH from a common level used in Dutch public indoor places (3 ACH, red line in Fig 4A) [61] to the recommended 6 ACH causes moderate effects and would reduce the exposure by aerosols with 20% (with the total exposure reduced from 81% to 65%).

Under the assumption that face masks block most droplets, the aerosols route becomes the dominant source of virus exposure (99%), even at low filter efficiency (FE) (Fig 4B). Assuming 40% FE for aerosols (i.e., 60% of aerosols and 6% of droplets pass through the face masks) [62], masks reduce the overall exposure to 28% compared to exposure without masks (red line in Fig 4B). The near-perfect protection at the highest FE (>75%) can be attributed to the additive effect resulting from mask-wearing by both infectious and susceptible individuals, provided the masks are well used and fitted [62,63]. Effectiveness would differ upon longer exposure or in settings with poorer ventilation, for instance.

How these intervention-induced reductions in exposure relate to infection risks is not straightforward and critically depends on the dose-response relationships of the several transmission routes. The impact of dose-response parameters on infection risks are further explored in a sensitivity analysis in section 3.2.3.

## 3.2. PeDViS application on a case study: simulating virus transmission at restaurants

We demonstrate the use of PeDViS with a case study, namely the simulation of virus spread and exposure in a restaurant setting. For the case study, a small conceptual restaurant is

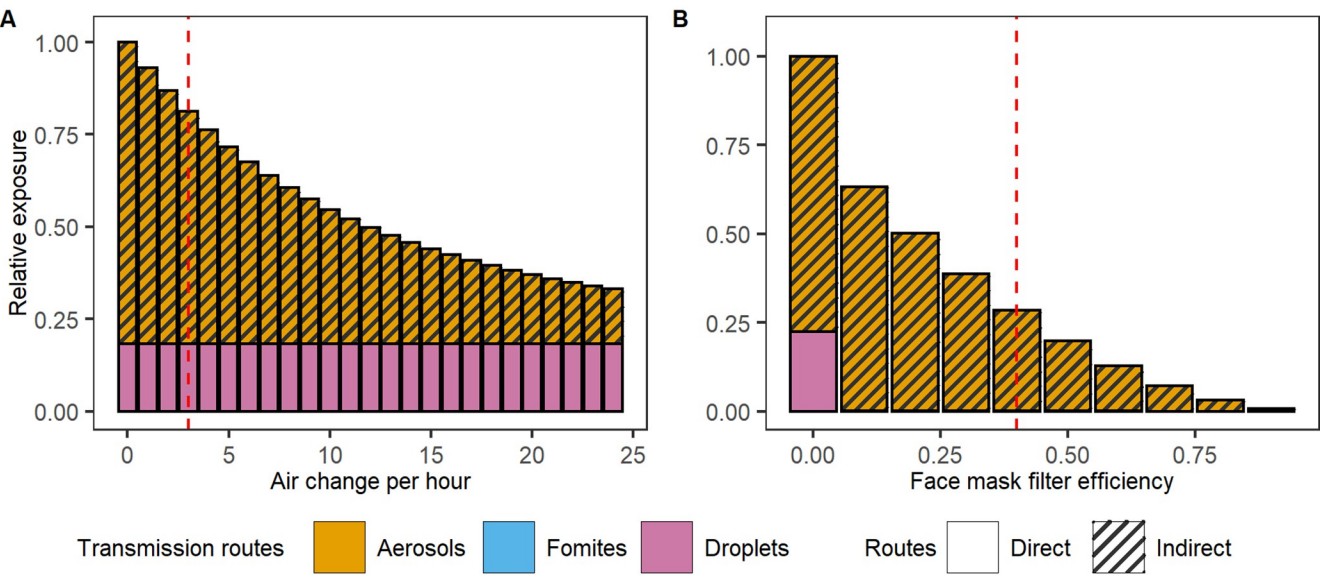

**Fig 4. The impact of interventions on exposure after 15 minutes at a 1.5-metre distance.** A) The impact of ventilation air change rate per hour (ACH) on exposure. The red dashed line shows the baseline ACH = 3 per hour indoors. B) The impact of mask-wearing by both infectious and susceptible individuals on virus exposure. The default filter efficiency is assumed to be 40% for aerosols (red dashed line). The exposure load for contact at 1.5m and 15 min without a mask and with poor ventilation (ACH = 0) is standardised to 1.

adopted, which has four tables, a bar and seating capacity of twenty people (Fig 5 and Fig B in S1 Text). The simulation lasts for 6 hours of service at a restaurant, in which some tables are used twice, and thirty two individuals in total enter the space. Only one infectious individual enters the simulation during its runtime, which is assigned at the beginning. The details of the case study setting are described in Section D in S1 Text.

**3.2.3. Results of PeDViS simulation of restaurant case study.** A case study in a restaurant was provided to show how human interactions drive transmission outcomes. The model simulated virus exposure of individuals in the restaurant and the impacts of face masks and ventilation thereon. In a sensitivity analysis we explored different dose-response relationships to estimate the number of infected individuals and the relative contribution of transmission routes.

In this section, first, the pedestrian movement dynamics are briefly discussed. Based on simulated movement trajectories, we present the viral spread through the restaurant's environment. The exposure to the virus for each of the individuals is then estimated. A sensitivity analysis on the relation between infection risks and virus exposure is done to align simulated infection risks to literature. Lastly, we evaluate the impact of interventions on reducing infections depending on the relative dose-response relationships assumed.

*Pedestrian movement dynamics in a restaurant setting.* To examine how human movement influences the exposure and transmission indoors, PeDViS was used to simulate a real-life scenario. The individual trajectories of one run with PeDViS are shown in Fig 5. Due to the stochastic activity scheduler and randomly drawn characteristics of the individuals, each run with PeDViS results in different trajectories. In order to fully comprehend the impact of infectious pedestrians in one space, one has to consider multiple runs with PeDViS, the exact number depending on the setting, occupancy, and the amount of distinct activities individuals engage in.

In the particular case visualised in Fig 5, the infectious individual (Individual 9) spent about 2 hours in total in this restaurant. It entered and sat at the middle table of the restaurant for 70

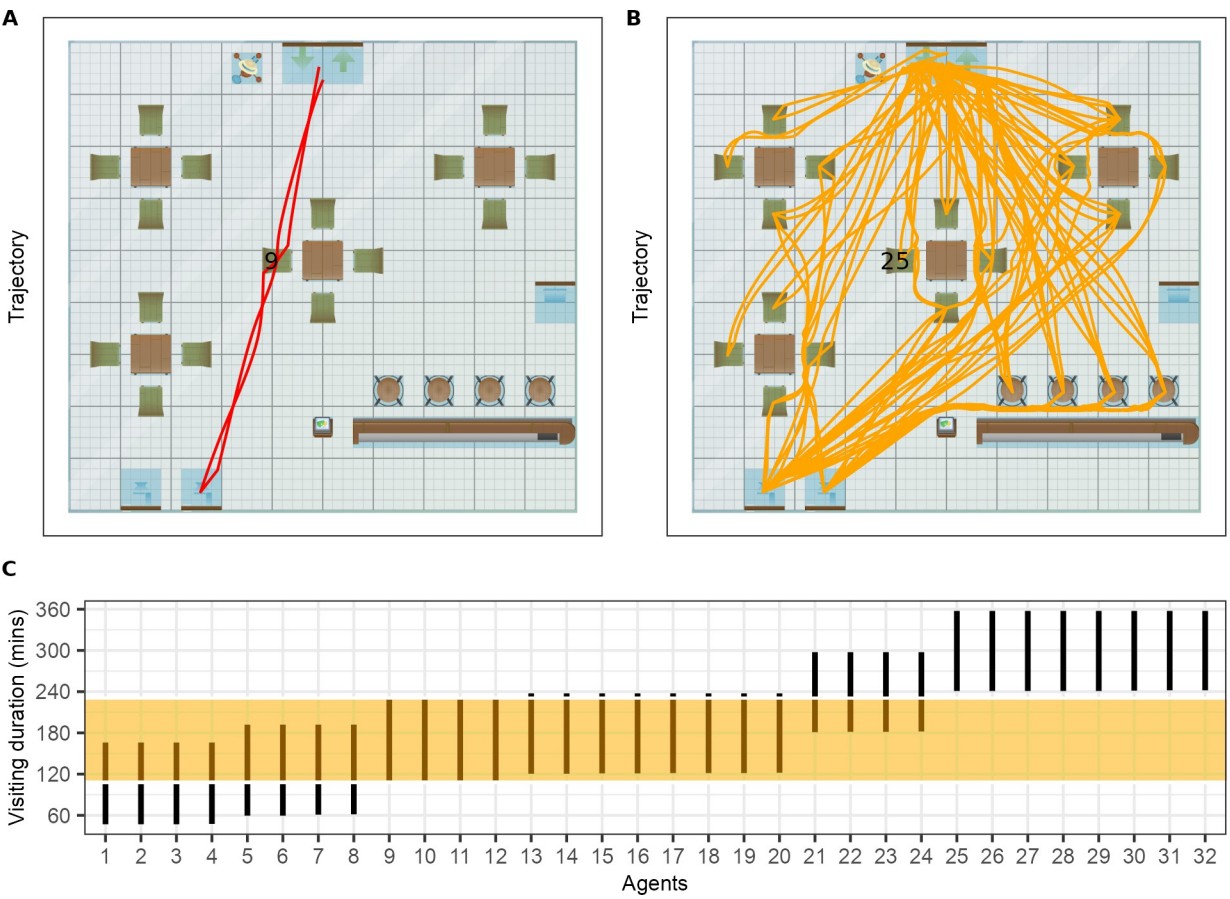

**Fig 5. The trajectory, seat locations and the visiting duration of each individual in a simulation.** (A) the trajectory of the infectious individual (ID = 9). (B) the trajectories of other 31 individuals with individual 25 sits in the same seat as Individual 9. (C) the visiting time of all individuals with the orange shade shows the visiting time of the infectious individual.

mins together with individuals 10, 11, and 12 (Fig 5C). Subsequently, Individual 9 went to the toilet for 4 mins and went back to their seat. Forty minutes later, individual 9 left the restaurant. As one can see Fig 5A, the trajectories of Individual 9 are relatively straightforward and direct. Individual 9 has spent most of its time sitting or standing at static locations. Twenty-three individuals walked into the space before or after Individual 9 and spent part of their time in the same room as Individual 9 (Fig 5C). Eight individuals entered the space after Individual 9 had left and did not have any direct contact with Individual 9. Individual 25 sat at the same seat as Individual 9 after it left (Fig 5B). Other than the others at the same table as Individual 9, most other individuals have not come into close vicinity for an extended period of time with Individual 9 during their stay. The main corridor between the entrance and the toilet is highly frequented, as is the route between the table on the right and the toilet.

*Viral spread.* The infectious individual's whereabouts determines the virus distribution in the air and on fomites (Fig 6). The cumulative contamination in heatmaps represent the accrued virus contamination. The contamination load is represented as a relative value as the amount of virus that an average infectious individual emits per hour with 30 mins breathing and 30 mins talking activity is standardised to 1 unit. The three maps illustrate that the contamination is highest near the chair where the infectious individual spends most of its time. This is, as expected, particularly clear in droplets (Fig 6B) and fomites (Fig 6C) heatmaps.

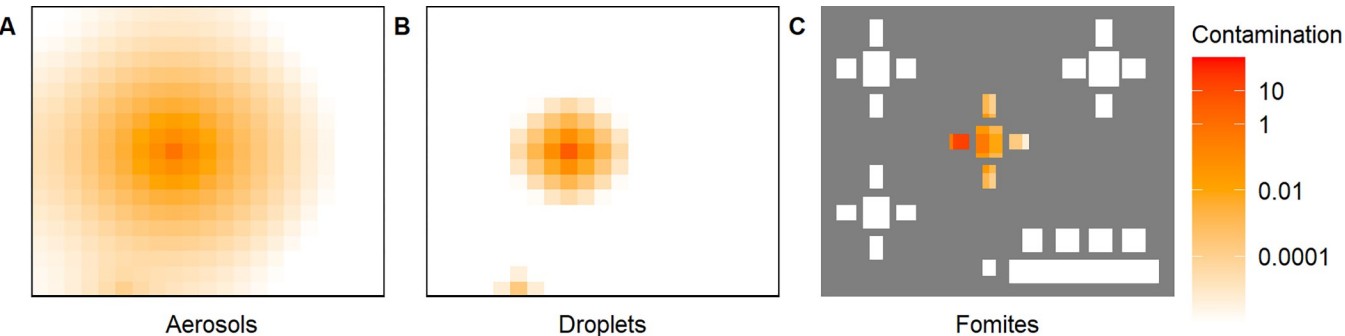

**Fig 6. Cumulative virus contamination in the environment.** (A) aerosols, (B) droplets, and (C) on fomites. Contamination is expressed as the virion quantity relative to an average infectious individual's hourly emission.

While the changes in the concentration of virus in droplets over time is tightly linked to the presence of the infectious individual, aerosols and fomites can build up in their environment and may persist after the infectious person has left (see snapshot heatmaps in Fig C in S1 Text).

*Individuals' exposure to virus.* The cumulative exposure of thirteen individuals (i.e., Individuals 10–21, 25) surpassed that of the benchmark contact (15 min at 1.5m), despite the fact that eleven of those individuals (all but 11 and 12) were never within 1.5m of the infectious individual (Fig 7). These thirteen individuals sat close to the infectious individual and overlapped sufficiently in time to get exposed to the virus or sat at the seat of Individual 9 after it left. Only the nearest neighbours (10 to 12) were exposed through droplet spread. Others were predominantly exposed through indirect routes, mainly aerosols. Only Individual 25 who sits in the same seat that the infectious individual (9) had occupied has been exposed to fomite as they shared common surfaces.

*Uncertainty relationship between virus exposure and risk of infection.* An individual's cumulative virus exposure is indicative of someone's risk of becoming infected, although the exact relation and how this differs by exposure route is uncertain. We applied exponential dose-response models, where the dose-response parameter $k$ for each route determines the number of virions someone is exposed to that results in a 63% probability of getting infected (see section 5.3.5). The value of $k$ varies between transmission routes due to different deposition location (eg. upper and lower respiratory tract) and deposition efficiency [64]. It is generally difficult to quantify $k$ by experiments [65–67]. Molecular epidemiological studies estimated bottleneck estimates to be around 1000 ($D_{inf}$) [68]. We treat this as a lower limit for $k$, considering that virions that contribute to an individual's exposure load, still need to overcome several barriers prior to reaching the cells of the respiratory tract ($c_{route}$). We performed sensitivity analyses by assessing the number of newly infected individuals expected to arise in this case study, assuming a range of proportional differences between the three routes ($c_{aerosols}$, $c_{droplets}$, $c_{fomites}$), and assuming an average infectious person emits $10^6$ viral particles per hour ($\phi$) when spending half of its time breathing and half of its time speaking (see details in Section B in S1 Text). This sensitivity analysis also captures the uncertainty around bottleneck estimates, which may well be tighter than 1000 [69–73]. The latter generally does not affect the results, as $c$ and $\phi$ scale linearly to exposure, with $c$ used to tune the results to epidemiologically reasonable outcomes. Substantial uncertainty in $\phi$ and $k$ should be considered when interpreting the estimates of $c$.

The number of infected individuals arising from this restaurant is most sensitive to the efficiency of aerosol transmission, with the mean number of infected individuals varying from 5.4

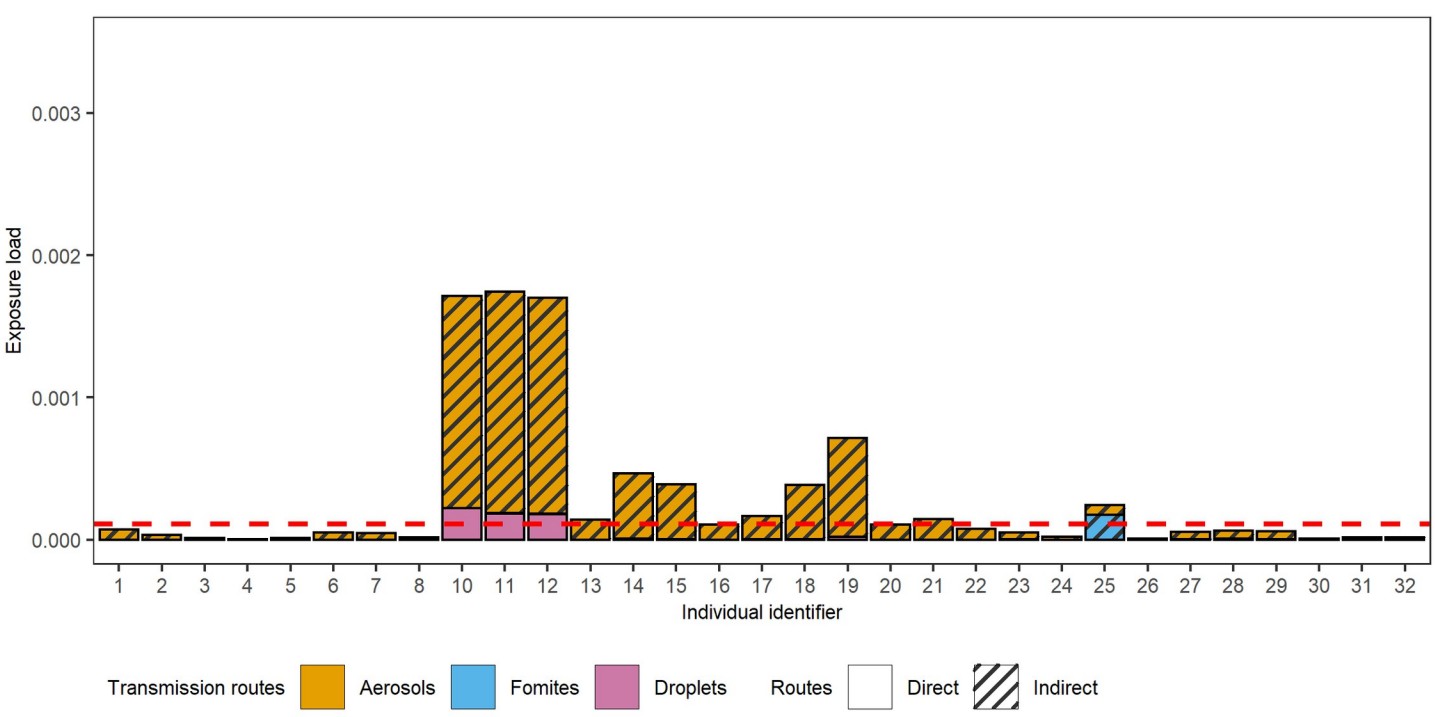

**Fig 7. Susceptible individuals' exposure load.** Exposure load is expressed as the virion quantity relative to an average infectious individual's hourly emission and is partitioned by transmission route. The exposure of susceptible individuals with the red dashed line showing the exposure for a benchmark contact of 1.5m for 15min.

with $c_{aerosols}$ at 100% to 0.02 when $c_{aerosols}$ is 0.1% (Table 1). The efficiency of the fomite transmission route ($c_{fomites}$) has little impact on the number of infected individuals in this specific case study due to limited sharing of surfaces between individuals. We considered a mean of 0.8 infections as a default, plausible scenario, in agreement with outbreak clusters data in similar social settings (mean secondary infections was 0.8, under the assumption that pairs with no

**Table 1. Sensitivity analysis on the impact of route specific dose-response relationships on the number of infected individuals (default in bold).**

| $D_{inf}$ | $c_{aerosols}$: $c_{droplets}$: $c_{fomites}$ | 5th percentile | mean | 95th percentile |
|---|---|---|---|---|
| 1,000 | 100%: 100%: 100% | 3 | 5.4 | 8 |
|  | 100%: 100%: 10% | 3 | 5.3 | 8 |
|  | 100%: 10%: 100% | 3 | 5.3 | 8 |
|  | 10%: 100%: 100% | 0 | 1.4 | 3 |
|  | 10%: 100%: 10% | 0 | 1.3 | 3 |
|  | 10%: 10%: 100% | 1 | 1.0 | 3 |
|  | 100%: 10%: 10% | 3 | 5.1 | 8 |
|  | **10%:10%10%** | **0** | **0.81** | **2** |
|  | 10%: 10%: 1% | 0 | 0.80 | 2 |
|  | 10%: 1%: 10% | 0 | 0.76 | 2 |
|  | 1%: 10%: 10% | 0 | 0.15 | 1 |
|  | 1%: 10%: 1% | 0 | 0.14 | 1 |
|  | 1%: 1%: 10% | 0 | 0.10 | 1 |
|  | 10%: 1%: 1% | 0 | 0.75 | 2 |
|  | 1%: 1%: 1% | 0 | 0.09 | 1 |
|  | 1%: 1%: 0.1% | 0 | 0.08 | 1 |
|  | 1%: 0.1%: 1% | 0 | 0.08 | 1 |
|  | 0.1%: 1%: 1% | 0 | 0.02 | 0 |

setting reported were proportionally distributed over the settings) [74]. As our default we use the efficiency estimates that give the best agreement with these empirical outcomes, which is when the most efficient exposure route has a $c_{route}$ is no larger than 10% (Table 1). We further adopt equal efficiency between routes ($c_{aerosols}$ = 10%, $c_{droplets}$ = 10%, $c_{fomites}$ = 10%) for our default dose-response relationship. Going forward, these route efficiency relationships are assumed, unless stated otherwise. The sensitivity of our model results to these assumptions is presented in the final part of this section, 3.2.3.

*Impact of interventions on exposure and infections.* Intervention measures differ in the transmission routes that they predominantly target. Here, as an illustrative example, we investigated how the combined effect of ventilation and face masks can reduce the distribution of virus in indoor spaces, the exposure of susceptible individuals, and ultimately the number of infected individuals. We compared five scenarios: (A) a 'worst case' scenario in which no interventions are applied and ventilation is poor (ACH = 0), (B) a baseline scenario with no interventions and with typical ventilation (ACH = 3), (C) with no interventions and with ventilation at recommended levels (ACH = 6) [75], (D) like (B) but with individuals wearing face masks when walking into and through the restaurant, and (E) like (D) but with increased ventilation (ACH = 6).

In a poorly ventilated restaurant (ACH = 0), the virus-laden aerosol concentration becomes higher than the baseline scenario (ACH = 3) (Fig DAa and Fig DBa in S1 Text). This increased aerosol concentration is sufficient to expose more people to the virus: thirteen additional individuals had exposures higher than that of a benchmark contact (IDs 1, 6, 7, 22–32) and only 5 individuals had exposure lower than a benchmark contact (ID 2–5, 8) (Fig 8A). The mean number of infected individuals in a poorly ventilated indoor space is estimated to be 1.59 times higher than in our baseline scenario (2.1 vs 0.81) (Fig 9Aa and 9Ab). Increasing ventilation to Dutch government recommendations (ACH = 6), the virus-laden aerosol contamination is reduced compared to the baseline scenario (Fig DCa in S1 Text), resulting in an estimated 41% reduction in the mean number of infected individuals (0.48 vs 0.81) (Fig 9Ac and 9Ab). There is a 61% chance of zero individuals getting infected, compared to 42% under the baseline scenario.

With a mere 1.2% reduction in infections (0.80 vs 0.81) relative to the baseline scenario, the impact of face masks was negligible (Fig 8B and 8D). This is due to the assumption that face masks are only worn while walking, as per Dutch guidelines that were in place. As a consequence, the only location where face masks have a notable effect on exposure is near the bathroom, and particularly for droplet spread (Fig DBb and Fig DDb in S1 Text). However, in this scenario, the risk of infection is low in these locations due to the short time people spend there. Including face masks to a scenario with increased ventilation has a similar effect, with an estimated 2.9% reduction in the estimated mean number of infections. Indeed, the impact of both interventions is compounded, owing to the different pathways that ventilation and face masks act on (aerosol vs droplet spread respectively).

*Sensitivity to route-specific infection efficiency.* Here we examine the impact of different assumptions on the dose-response curves on the impact of interventions. Specifically, we considered the infectious dose ($D_{inf}$) and its relation to the average emission rate known and vary the proportion of virions someone is exposed to reaching the cells of the respiratory tract target cells ($c_{route}$) (Fig 9). We consider four scenarios: i) virions have equal probability of reaching the respiratory tract target cells, irrespective of the exposure route, ii) like i but virions that someone is exposed to through fomites have lower $c$ iii) like ii but with droplets having a lower $c$ than aerosols, and iv) like ii but with aerosols having a lower $c$ than droplets. We examined the mean number of infections that may have arisen from the described case study.

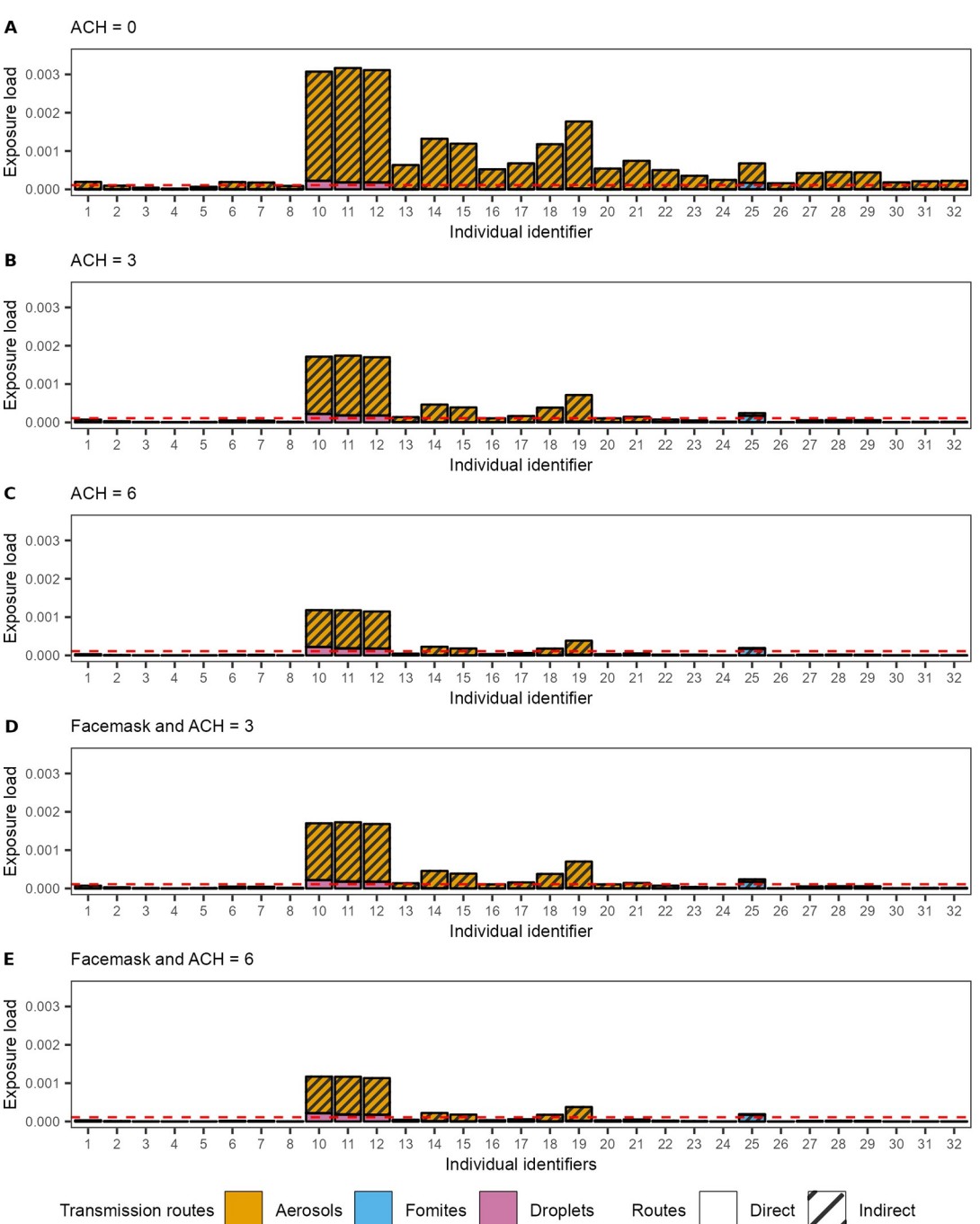

**Fig 8. The impact of face masks and ventilation on virus exposure in the case study.** (A,B,C) a scenario where individuals do not wear face masks and an ACH is 0 (A), 3 (B), and 6 (C) per hour in the restaurant, (D, E) a scenario where people wear face masks when moving and an ACH of 3 (D) and 6 (E). The dashed red line indicates the expected exposure of a benchmark contact of 1.5m for 15 minutes.

Considering the baseline scenarios (Fig 9Ab–9Db) of no intervention (i.e., no face masks) and average ventilation (ACH = 3), the mean number of infections ranges from 0.81 to 0.1, depending on assumptions on the relative transmission efficiencies of the different routes. Infection estimates are lowest when aerosol spread is assumed less efficient (mean = 0.1, 87.5%

 

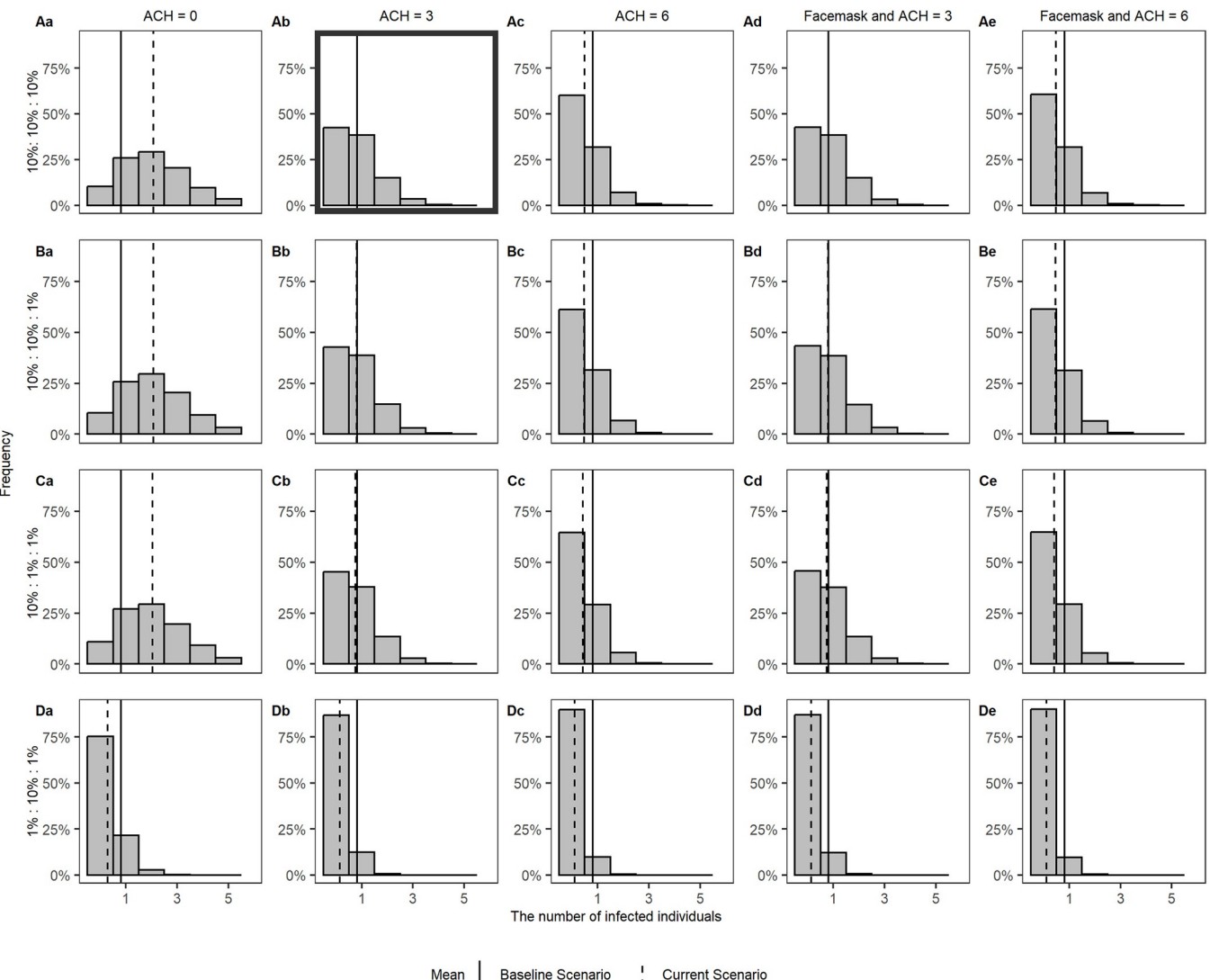

**Fig 9. The density distributions of the expected number of infected individuals in the case study for varying route-specific transmission efficiency.** Each row shows a parameter setting for $c_{route}$: (A) $c_{route}$ is the same for all routes ($c_{aerosols}$:$c_{droplets}$:$c_{fomites}$ is 10%:10%:10%). (B) $c_{route}$ is smaller for fomites ($c_{aerosols}$:$c_{droplets}$:$c_{fomites}$ is 10%:10%:1%). (C) $c_{route}$ is smaller for fomites and droplets ($c_{aerosols}$:$c_{droplets}$:$c_{fomites}$ is 10%:1%:1%). (D) $c_{route}$ is smaller for fomites and aerosols ($c_{aerosols}$:$c_{droplets}$:$c_{fomites}$ is 1%:10%:1%). Each column shows an intervention scenario: (a) poor ventilation scenario, ACH = 0, (b) baseline scenario, ACH = 3, (c) scenario with recommended ventilation, ACH = 6, (d) baseline scenario with face masks worn while moving, (e) scenario with recommended ventilation and with face masks worn while moving. The black solid lines indicate the mean value of the infected number in the baseline scenario and the dashed lines show the mean value corresponding to each respective intervention scenario. Fig 9Ab (in bold border) shows the baseline scenario.

reduction relative to the default of ($c_{aerosols}$ = 10%, $c_{droplets}$ = 10%, $c_{fomites}$ = 10%) (Fig 9Db). Assuming less efficient transmission through fomites or through fomites and droplet exposure results in a smaller reduction (mean = 0.79 or 0.74, 2.7% or 8% reduction relative to the default) (Fig 9Bb and 9Cb). Due to the wider spatial distribution of aerosols, more individuals get exposed through this route. The total number of infections is therefore most sensitive to the aerosol specific dose-response relationship. Whereas aerosols (short and long range) would be accountable for 90% of infections under the default assumption ($c_{aerosols}$ = 10%) (Fig EAb in S1 Text), this is reduced to 55% if aerosol transmission is assumed less efficient ($c_{aerosols}$ = 1%) (Fig EDb in S1 Text).

The sensitivity on the assumed dose-response relationship further becomes apparent when comparing the impact of ventilation on the estimated number of infections. Whereas, under default assumptions and relative to average ventilation (ACH = 3), poor ventilation (ACH = 0) would be associated with a 2.44-fold increase in the number of infections, (Fig 9Aa and 9Ab), this difference would be diminished if virions in aerosols would infect an order of magnitude less efficiently than those in droplets. (Fig 9Db and 9Dc).

Since the use of face masks while walking was not found to substantially affect individuals' virus exposure, the total number of infections averted is less sensitive to assumptions on the dose-response relationships. The largest impact is seen in a scenario in which droplet spread is the most efficient route of transmission (Fig 9Cb and 9Cd).

## 4. Discussion

Although SARS-CoV-2 continues to circulate at high levels around the world, COVID-19 is no longer considered a global health emergency. Experiences from the COVID-19 pandemic are now being used to inform response plans for future pandemics by virulent, immune-escaping SARS-CoV-2 variants or other pathogens. Interventions targeted at reducing transmission in indoor spaces will constitute an important part of these plans, particularly for pathogens for which no pharmaceutical interventions are (yet) available. Here, we presented the hybrid simulation model PeDViS, a tool that can contribute to the improved understanding of indoor transmission and guide preparedness efforts. It simulates the interplay between pedestrian's choice and movement dynamics, in the specific context of indoor spaces, and the spread of respiratory viruses. We introduced this new model framework and demonstrated its use in identifying where and when at-risk contacts occur in real life scenarios in indoor spaces. We illustrate how this information can be used to inform intervention measures, and demonstrate that the impact of combined intervention strategies crucially depends on the efficiency of distinct transmission routes.

Many interventions in indoor spaces aim at reducing the number of proximate contacts visitors have. However, not all proximate contacts constitute a real risk for transmission. We aimed to get a better understanding of what constitutes a risky contact and how this differs depending on the setting in which this contact takes place. The explicit modelling of the spatial distributions of virions in the environment allowed for the exploration of how virus exposure may relate to the duration and distance of potentially infectious contacts. Specifically, we simulated an exponential decay in virus exposure over distance, with little exposure beyond the commonly used benchmark of 1.5m, provided the contact is of short duration. Longer contact durations are expected to be associated with buildup of virus in the environment, increasing virus exposure, also beyond 1.5m. The buildup of virus in environments can further contribute to elevated virus exposure when an infectious person has spent a substantial amount of time in that same space, before the contact takes place. Whether and how often such indirect transmission events occur, is hard to verify from epidemiological surveillance data, but has been demonstrated to be possible in animal experiments [76].

We used PeDViS to assess the frequency and intensity of contacts that take place in a specific setting, based on the activities performed in a space and typical pedestrian dynamics (i.e., as they arise from route choices and collision avoidance). This part of the modelling relies on the well-established pedestrian model NOMAD, which has been updated for this work for use in small-scale settings. While it allows for the inclusion of physical distancing, crowd monitoring data gave little support for a substantial effect of distancing methods and were therefore not included here. The NOMAD model gives the PeDViS framework the ability to construct contact networks for a wide range of settings and, through pairing with QVEmod, tie this to

exposure risks. These exposure risks cannot be easily related to a single benchmark contact (here within 1.5m for at least 15 minutes), due to the intricacies of indirect, airborne transmission. For instance, the case study shows that, of the thirteen individuals whose cumulative exposure surpassed that of a benchmark contact, ten had never been within 1.5m of the infectious individual. Yet, their visits had overlapped sufficiently in time with the infectious individual to accrue virus that had built up and was distributed in the environment. Modelling efforts, such as the ones performed by PeDViS, can help assess the added risks associated with such indirect exposure routes (i.e., aerosols and fomites) by accounting for the impact of individuals sharing spaces, even if not (entirely) concurrent in time.

We examined the relative reductions in virus exposure that results from different intervention measures and showed that the impact of these measures may well be context specific. While in poorly ventilated spaces, by increasing ventilation to an average level, great reductions in virus exposure can be achieved, increasing ventilation beyond this level has a smaller accrued effect. Similarly, face masks by the guests likely have little impact if not worn while seated, as this is when longer, static contacts occur. However, one incentive of such masking orders is to reduce the risk of contacts with individuals outside of one's own social circle (i.e., those not seated at the same table). For the restaurant setting explored, we postulate that the encounters whilst walking to and from one's dining table are sufficiently short to pose a minor risk to other people. The role of masking of personnel was not explored in this study but is expected to be more effective due to the frequent encounters they have with guests and colleagues and the long duration they spend in the space. Future iterations of the model will include the additional activity models for personnel required for examining this question.

How the route specific exposure to the virus relates to infection risks remains an open question [65,67]. This question both relates to the challenges involved with investigating and quantifying the biological processes in laboratory settings as well as the need for model validation based on epidemiological data. Beyond the challenges of estimating and validating the emission and spread of viruses in environments, empirically measuring the rates at which virus is inhaled and/or picked up and subsequently reaches the respiratory tract target cells typically relies on indirect estimations [77,78]. Subsequently, as different target cells present different populations of receptors [79], the infection success of a virion may well depend on where in the respiratory tract it deposits. The mucous layer also likely differs in terms of permeability and clearance mechanisms across the respiratory tract [80]. We captured these different levels of uncertainty and variability in a single parameter $c$, which determines what proportion of virions, after exposure, successfully reaches the respiratory tract target cells [68,81]. The order of magnitude was scaled such that the distribution of cases matches that of a large infector-pair study in restaurant settings [74]. While this was not intended as a formal calibration, it should result in a rough ballpark estimation that harbours realistic numbers of infections. For this and other indoor transmission models, future efforts should include formal validation exercises that assure that the emerging properties of from the bottom up parameterized modelling systems align with fine-scaled epidemiological outbreak data.

The main purpose of this effort, examining the relative impact of intervention measures, is particularly sensitive to the assumed magnitudes and differences in transmission efficiency between routes (Fig E in S1 Text). In particular, the uncertainties in the efficiency of aerosol transmission affect the impact of interventions. As aerosols can both disperse and accumulate over time, they may contribute to transmission over distances longer than 1.5m, especially if the infectious person is present in the space for a prolonged duration. Superspreading events associated with poorly ventilated spaces are indicative of a role for aerosols in transmission [8,82–86]. The extent to which aerosols contribute to transmission in spaces with adequate ventilation depends on the efficiency of this route (Fig E in S1 Text) and will differ between

settings [78,87]. Similarly, the case study examined did not present a large contribution of fomites to transmission. In many infectious diseases, particularly those whose transmission through surfaces plays a major role such as Ebola or chicken pox, shared surfaces can be an important infection transmission route. PeDViS can simulate the virus transmission mechanism through surfaces along with the aerosols and droplets and is thereby generic in representing all transmission routes relevant to respiratory pathogens. However, for COVID-19, evidence shows that the virus mainly spreads through respiration [88–90], and transmission through surfaces may be limited [91–95]. It can, however, not be ruled out and might play a role in specific scenarios. Experimental studies in cats, for example, have shown that SARS-CoV-2 can be transmitted through the environment [76]. This transmission is primarily associated with the accumulation of the virus in the environment over prolonged time in a shared space rather than being linked to high-touch surfaces. Alternatively, simulation studies [96] illustrate that high-touch surfaces could potentially play a role in crowded settings such as train carriages, where some surfaces are potentially shared by many different individuals. For this particular case study, this low probability transmission route is considered only through the main activity areas of the customers, which are their tables and chairs. However, other scenarios with conditions more favourable to fomite transmission, including the possible transmission through other high-touch surfaces such as door handles, coat rack or pay register, can be examined to better understand the potential for contribution by this route, for SARS-CoV-2 and other pathogens. For instance, the study on controlled transmission in cats enumerated that, in that specific scenario, one third of transmission could be attributed to indirect, environmental transmission [76], highlighting that, albeit not the major source of transmission, SARS-CoV-2 has the potential to be transmitted through fomites.

There are limitations to this study. Some parameters are hard to quantify empirically, are setting-specific, and/or vary greatly between individuals. For others, data is too sparse to draw strong conclusions. The model presents what we believe to be the currently available empirical evidence and shall be updated whenever new, valuable data become available. It can further be adapted to reflect different variants. While many of the model parameters may affect the absolute virus exposure, predicted infection risks were found to be robust to changes in most of the parameters explored (see details in Figs F-L in S1 Text). Infection risks are most sensitive to different levels of emission rates. Here, these are assumed proportional to individual viral loads, which are known to be highly heterogeneous, both between individuals as over the course of the infection [97]. Heterogeneity in infectiousness may, among other factors, be an important source of heterogeneity in observed outbreak sizes. In a sensitivity analysis on the emission rates, the average number of secondary infections varied from 0.09 to 5.40 (Fig F in S1 Text), reflecting that, while most individuals will on average not contribute to onward transmission, some may affect many [74]. PeDViS can be used to further disentangle the sources of heterogeneity that together result in the highly overdispersed outbreak sizes observed for this pathogen. Further, the division between droplets and aerosols is somewhat arbitrary [98]. We used the conventional discrete cut-off size to classify droplets and aerosols (d = 10um), so as to align with the definitions in public health guidance [98]. In addition, the airflow (i.e., the diffusion of air) is modelled to be homogeneous across the space and follow the same mechanism in all directions. Hence, the diffusion rate is independent of any external effects (e.g., temperature, ventilation, space occupancy). This simplification is intentional and should provide generic results. However, more directed airflows could alter transmission risks by resulting in increased exposure in some places and reductions in others. In future efforts, this model will be paired with more detailed airflow models.

In this specific exercise, we did not present the full expected variation in outcomes but rather demonstrated the model application here with a single NOMAD replication of our case

study restaurant. Both the movements of the guests and the infectiousness of the infected person were identical between runs, as was the assignment of the infectious person (who is always seated at the middle table). As such, the simulation experiments could be regarded as a repetition of a single evening in a restaurant that takes place under a select set of scenarios (Fig 5). This allowed us to make direct comparisons between runs and single out the impact of interventions or uncertainty in parameter values. While these specific runs thus do not account for the several sources of stochasticity that underlie the indoor transmission events, the model and accompanying application are set up to do so. One can readily expand the types and configurations of restaurants and compare findings over large sets of iterations including several sources of randomness. For instance: the activity scheduling and NOMAD sections of the model simulate randomness in guests' entrance and leave times, walking speed, and the probability of visiting the toilet. In QVEmod, the assignment of infectious agents is randomised as well as whether a specific virus exposure results in infection. Further, sources of individual-level heterogeneity, such as in infectiousness and respiratory activities, can be examined towards a better understanding of the drivers of superspreading events. Lastly, in current simulations, only guests to the restaurant are simulated. Guests have rather similar activity schedules when visiting a restaurant, resulting in a relatively easy, tangible example in which the index case is among guests, which are mostly stationary. The numerous short range contacts made by potentially infected personnel and the longer time spent in a space will result in different dynamics of spread and consequently a different set of interventions. Next iterations of the model will aim to address these questions.

Many intervention measures applied during the pandemic relied on behavioural changes in response to non-pharmaceutical interventions (NPIs) that aim to reduce infectious contacts in public indoor spaces. The population-level impact of such measures depends on the contribution of specific settings to overall transmission, which follows from i) the time people spend in specific settings and ii) the by-setting risk for an infected individual to infect other people while there. PeDViS is developed to help inform the latter. The use of fine scale pedestrian modelling allows for the characterisation of the human interactions that emerge in various indoor settings. It is the frequency and intensity of these interactions, coupled with the environmental factors that affect the efficiency of transmission, that determines the setting-specific risk of transmission. Here, we worked with an estimate of on average 0.81 infections arising from the infectious individual, in line with empirical estimates [74]. This estimate should be regarded as one component of the individual reproduction number, as it denotes the number of new infections caused by a specific infected individual during part of its infectious period. The full estimate being derived from adding up the infections estimated to arise from each setting visited over the course of one's infectious period. The reproduction number for the population can be derived from the individual reproduction numbers, while accounting for the individual-level probabilities of getting infected. To reduce population-level transmission, intervention measures focused at indoor spaces should aim to reduce the reproduction number to below one, by either reducing time spent in spaces with high by-setting transmission risk or by reducing the risk in such settings. Here, one should also consider that the reason for visiting a setting could affect one's contribution. For instance, personnel are expected to have contact structures that are markedly different from guests. Also, personnel have a larger probability of visiting a restaurant setting multiple times during their infectious period. This could increase their importance to restaurant transmission and possibly to overall transmission. The latter also depends on their risk of acquiring infection, which could, due to having a profession with frequent proximate contacts, be higher than the general population. However, such quantifications would require a more complete understanding of how people spend their time before and over the course of their infectious period [99].

The PeDViS model can be readily adapted to different SARS-CoV-2 variants and respiratory viruses and to populations with different levels of immunity. Owing to the modular set up of PeD-ViS, it can be used to characterise the infection risks in other types of indoor spaces, with different human movement and behaviour characteristics, and with a wide range of possible interventions. While the uncertainties surrounding many of the model parameters limits the ability to estimate actual numbers of infections arising from a scenario, estimating relative changes in response to interventions is robust for most scenarios and can help guide public health decision making.

## 5. Methods

This section presents the details of modelling methodology. First, Section 5.1 details the high-level pedestrian activity choice behaviour models, which comprise of an activity, destination, and departure choice assignment models. Section 5.2 continues with a description of the operational movement model, in particular NOMAD. The last section (5.3) provides an overview of the virus spread and risk identification models that form the last part of the modelling chain.

### 5.1 Activity scheduler model

There is limited work featuring the modelling of activity choice behaviour in buildings (See more detailed literature review in Section A in S1 Text). Most activity assignment models are very specialised for certain types of buildings, predominantly offices or require extensive data. Thus, the authors have decided to develop a new pragmatic activity assignment model, in this case one specifically tailored to restaurants. The main design features of the new model are that it can create a variety of activity behaviours whilst requiring few and simple inputs. Below, the inputs and the model are further detailed.

**5.1.1. Activity scheduler inputs.**   Based on consultation with people in the restaurant industry a number of inputs have been identified. These inputs are a combination of those necessary for the model to create realistic activity patterns and those that can be easily and realistically provided by restaurant owners. The selected input are:

1. The restaurant layout: This includes the number of tables and number of chairs per table and their location, the location and amount of toilets (if they are present), the location of a coat rack (if present) and the location of a register (if present).

2. The time period that should be simulated.

3. The demand pattern: This input divides the overall time period into smaller time slots and for each of those defines how many groups will visit the restaurant during that time.

4. The expected average duration of guest visits.

Together these inputs provide the activity model with the information it needs to create the activity schedules for each individual guest.

**5.1.2. The activity choice and scheduling model.**   The activity choice and scheduling model uses a two-step approach to create the activity schedule of each individual guest. The first step involves scheduling the visit of all groups of guests. This step results in the start and end time of the visit of each group, the table to which they are assigned during their visit and the group size. The second step then creates an activity schedule for each individual of each group.

In the first step, the model first creates a provisional schedule that ensures that each group, which is scheduled to visit the restaurant according to the demand pattern, is assigned a table and a provisional start and end time. The start and end time are chosen such that:

• The start time of each group falls within the time slot provided by the demand pattern.

- The visit duration (the difference between the end and start time) is at least the expected average duration provided by the input.

- Any table is only occupied by one group at a time.

Next, it computes the actual start and end time of each group's visit by taking the provisional start and end time and adding some variation. This ensures that groups within the same time slot have slightly different visit durations and arrival times.

In the second step, the model takes the visit start time, the visit end time, and the group size of each group to create an activity schedule for each individual of the group. The schedule of each individual guest consists of a number of mandatory activities, some optional activities and some conditional activities. These are the following (in order):

- Enter the restaurant: This is always the first activity and a mandatory activity

- Hang coat at the coat rack: This is an optional activity performed after entering the restaurant provided a coat rack is available and the guest chooses to use it given a certain probability.

- Sit at the table: A mandatory activity performed after entering the restaurant or using the coat rack

- Go to the toilet: An optional activity provided a toilet is available and the guest chooses to use it given a certain probability. Afterwards the guest returns to the table.

- Pay at the register: A conditional activity assigned to only one member of a group provided the payment is not performed at the table.

- Pick up coat from the coat rack: A conditional activity provided the guest chose to hang their coat at the coat rack when entering the restaurant.

- Leave the restaurant: The last activity and a mandatory one.

All individuals of the same group will enter the restaurant at roughly the same time and will leave at the same time. By adapting the different probabilities and durations of the activities a range of activity schedules can be produced that fit different restaurants. For a more detailed description of the activity model, see [53].

## 5.2. Operational model—NOMAD

NOMAD is a microscopic simulation model that simulates the operational movement dynamics of individuals. In particular, the walker model is implemented in PeDViS (see Eqs 1–6). The result of NOMAD is a set of trajectories pertaining to the coordinates and velocity of each individual in the simulation at each timestep of the simulation.

**5.2.1 Routing model—NOMAD.**   The routing model of NOMAD is utility-based and developed by Hoogendoorn and Bovy [54] and makes use of the minimum walking cost principle. In essence, individuals balance their desire to move towards their destination with other needs, for instance travel time, physical effort, closeness to attractive sights. In this implementation of NOMAD, only the need to avoid static obstacles in their surroundings is accounted for. Using a floor field approach, the walking costs are computed for the complete walkable area of the pedestrian infrastructure. In particular, a grid of rectangular cells (0.1x0.1m) is adopted, each of which contains a cost value. Based on the static cost map, the desired direction of an individual in the centre point of each cell can be determined using the steepest descent method. Here, individuals are walking orthogonal to the equi-cost lines. A continuous representation of the desired direction can accordingly be calculated on the fly by means of

linear interpolation between the actual location of an individual and the four nearest locations for which the desired direction was already computed. See Fig 10 for an illustration of two trajectories that could be the result of this routing model.

**5.2.2. Operational dynamics—NOMAD.** Underneath, the main elements of this model are briefly introduced. For an in-depth discussion of the walker model and its calibration one is referred to [101].

$$\frac{d}{dt}\overrightarrow{r}_p(t) = \overrightarrow{v}_p(t) \tag{1}$$

$$\frac{d}{dt}\overrightarrow{v}_p(t) = \overrightarrow{a}(t) \tag{2}$$

$$\overrightarrow{a}(t) = \overrightarrow{a}_c(t) + \overrightarrow{a}_p(t) + \overrightarrow{\xi} \tag{3}$$

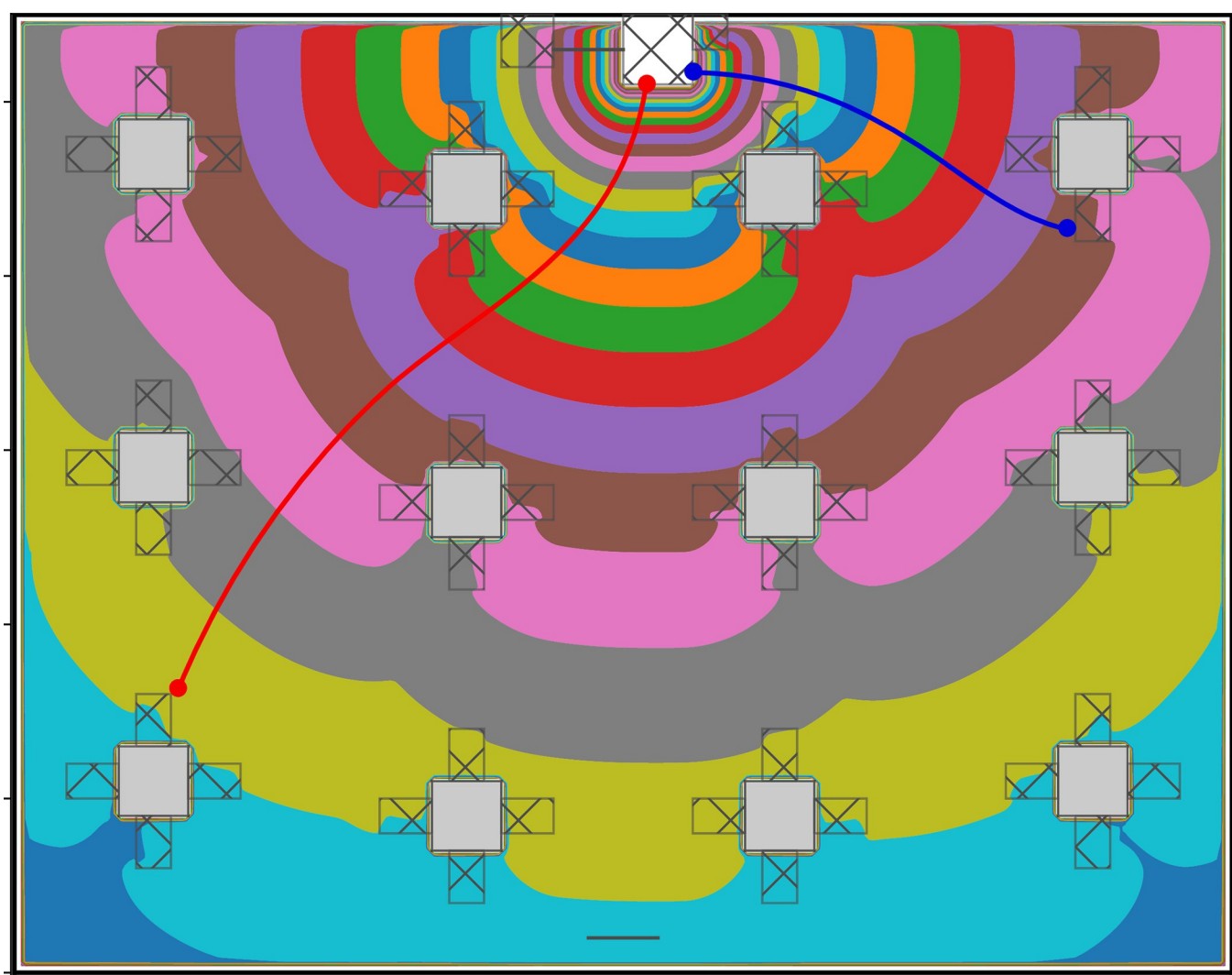

**Fig 10. Illustrative NOMAD floor field with two resulting trajectories ([100]).** The white rectangle (on the top) represents the entrance, grey rectangles represent the tables which also act as obstacles, the transparent rectangles around the tables are chairs that represent the destinations, coloured areas in the middle show how the walking cost fields are shaped over the space, and the red and blue lines are examples of the preferred path a pedestrian would follow.

$$\overrightarrow{a}_c(t) = \overrightarrow{a}_s(t) + \overrightarrow{a}_O(t) + \overrightarrow{a}_{pq}(t) \tag{4}$$

$$\overrightarrow{a}_s(t) = \frac{(v_0(t) \cdot \overrightarrow{e}_g) - \overrightarrow{v}(t)}{\tau} \tag{5}$$

$$\overrightarrow{a}_{pq}(t) = -\overrightarrow{e}_{pq} \cdot A_0 \cdot e^{\frac{-d_{pq}}{d_i}} \tag{6}$$

$$\overrightarrow{a}_O(t) = -\overrightarrow{e}_O \cdot A_O \sum_{o \in O} \begin{cases} 1 & \text{for } 0 < d_{pO} < d_0 \\ 1 - (d_{pO} - d_0) & \text{for } d_0 < d_{pO} < 2d_0 \\ 0 & \text{for } d_{pO} > 2d_0 \end{cases} \tag{7}$$

Within NOMAD, the movement of pedestrians is assumed to be accelerations that are caused by signals and forces that pedestrians are subjected to. These accelerations are partly controlled $\overrightarrow{a}_c(t)$ and partly uncontrolled $\overrightarrow{a}_p(t)$. A noise term $\overrightarrow{\xi}$ comprises the last part of the accelerations, which simulates the natural fluctuations of pedestrian movements. Together these three acceleration reactions shape the acceleration of an individual.

The controlled reaction $\overrightarrow{a}_c(t)$ is the result of the individual's desire for a certain velocity $\overrightarrow{v}_0(t)$ (i.e. speed and direction), the physical interaction with other pedestrians, and surrounding objects. Here, $\overrightarrow{a}_s(t)$ represents the path straying component, $\overrightarrow{a}_{pq}(t)$ the pedestrian interaction component and $\overrightarrow{a}_O(t)$ represents the obstacle interaction component (see Eqs 4–7). $A_O$, $A_0$, $d_i$, and $d_0$ represent parameters that respectively determine the strength of the pedestrian interaction and obstacle interaction forces. Please note, the parameters of NOMAD do not influence the movement dynamics of the simulated crowd to a similar extent, since not all forces are always present. Forces with respect to obstacles and pedestrians are only significant if the pedestrian resides within range of obstacles or pedestrians.

*Path straying*. When walking, individuals have a desired velocity (combination of speed and direction) that is aligned along the optimal route and speed towards the destination of the pedestrian. NOMAD assumes that deviations from the optimal speed and/or direction incur increasing costs. Therefore, pedestrians always attempt to return to their optimal velocity $\overrightarrow{v}_0(t)$. Tau represents the relaxation term, which identifies the desire of pedestrians to keep moving $\overrightarrow{e}_g$ towards their goal along their intended global path. The smaller $\tau$, the longer the time that individuals require to alter their speed and direction.

*Interaction with other pedestrians*. NOMAD models the collision avoidance behaviour by means of a non-cooperative game theory strategy [102]. Pedestrians minimise walking costs by anticipating the movement of others and themselves. Besides that, NOMAD's reaction to other pedestrians is anisotropic. That is, pedestrians have a limited ellipse area in which they interact with other pedestrians and obstacles. The interaction costs of an interaction between two individuals is the inverse of their heart-to-heart distance. Thus, the closer individuals are, the larger the collision avoidance forces, which are pointing in the direction opposite of the interaction. Here, $A_0$ identifies the interaction strength, $d_i$ interaction distance, $d_{pq}$ the anticipated distance and $\overrightarrow{e}_{pq}$ the unit vector pointing in the direction of the other pedestrian.

*Interactions with obstacles*. The strength of the interaction with obstacles is dependent on the distance to the obstacle $d_{pO}$, the interaction strength of objects in general $A_O$ and the direction of the nearest obstacle $\overrightarrow{e}_O$. Here, a step-based approach is used, where obstacles nearby

**Table 2. Parameters for activity scheduler and pedestrian model.**

| Attribute parameters | Value | Range |
|---|---|---|
| $\tau$ | 0.5 [s] | |
| $A_0$ | 2.0 [m/s2] | |
| $d_i$ | 0.4 [m] | |
| $A_O$ | 1.5 [m/s2] | |
| $d_0$ | 0.1 [m] | |
| Desired speed customers | N(0.9, 0.2) [m/s] | [0.4, 1.4] [m/s] |
| Pedestrian radius | 0.15 [m] | |
| Toilet visit probability | 0.6 [–] | |
| Toilet visit duration | N(120, 60) [s] | [100, 240] [s] |
| Coat rack visit duration | 20 [s] | |
| Register visit duration | N(30, 10) [s] | [20, 50] [s] |
| Pay at table duration | 60 [s] | |

have a very large influence and obstacles outside a range of influence $d_0$ not influence individuals' movement dynamics at all. Two distance thresholds ($d_0$ and $2d_0$) are used to govern the gradual linear decline of the obstacle avoidance force. As a result of the formulation, agents within NOMAD only react to obstacles when they are really close to the obstacle. This is an advantage in case of the modelling of indoor spaces, where lots of obstacles are present.

The parameter values used in NOMAD are depicted in Table 2.

**5.2.3 Parameters setting in NOMAD.** The output of NOMAD is detailed data on the movements and activities of all agents in the model. For each agent, the position is recorded every 0.1 seconds resulting in a detailed trajectory per agent. These outputs are converted into the inputs of the **Virus Spread Model:QVEmod** in the form of a script for each agent after the conversion of the time step from 0.1 seconds to a configurable user-defined value (default = 0.5 minutes).

## 5.3 Virus Spread Model: QVEmod

A spatially explicit agent-based model was developed that simulates emission of viruses by infectious individuals, how these subsequently spread in space and over time within an environment, and eventually may get picked up by susceptible individuals. The model distinguishes seven processes (Fig 11):

i.  An infectious individual emits virus into the air through virus-laden aerosols and virus-laden droplets (further referred to as aerosols and droplets, depending on their size).

ii.  Droplets deposit onto surfaces.

iii.  Viruses lose infectivity at a rate depending on their state in the environment (airborne or on surfaces).

iv.  Viruses in droplets and aerosols diffuse in the air.

v.  Susceptible individuals can get exposed to viruses through inhaling air with viral-laden droplets and aerosols.

vi.  The infectious individual contaminates surfaces by touching objects in the space (e.g., tables, chairs, and menus).

vii.  Susceptible individuals can be exposed to viruses by touching contaminated surfaces (fomites).

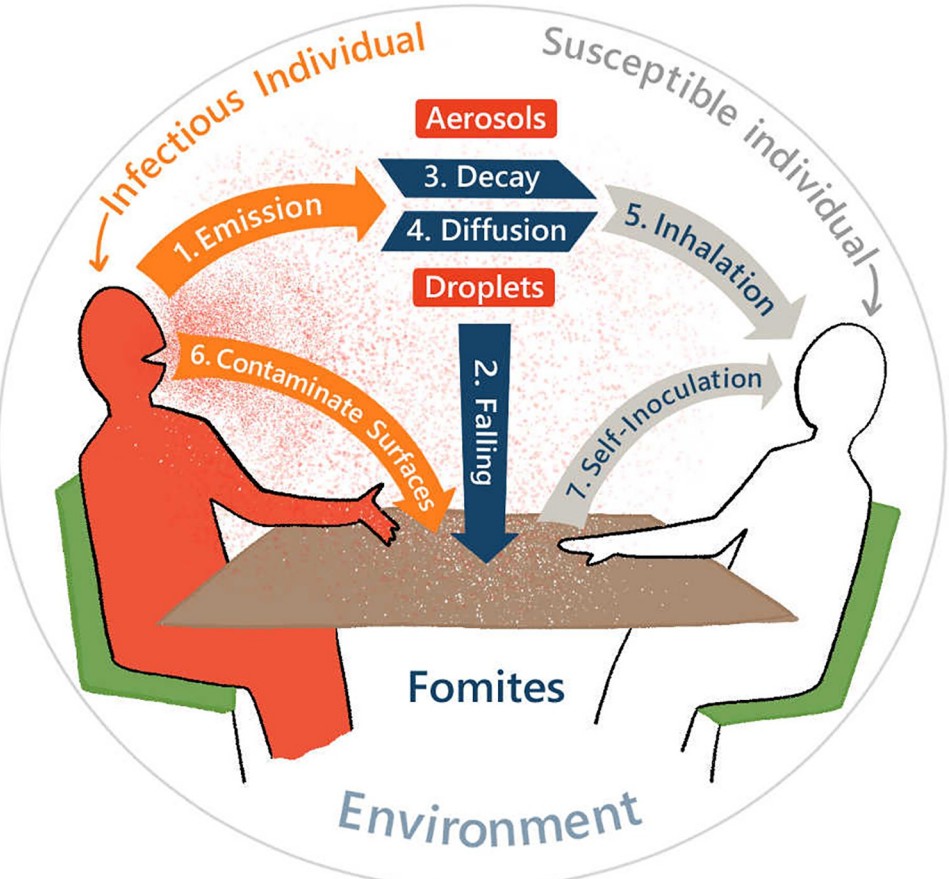

**Fig 11. Schematic of processes in the epidemiological model.**

A description of the state variables and initialization processes is provided in sections 5.3.1–5.3.2. The equations associated with the seven core processes and the parameterisation of the model are described in detail below in sections 5.3.3. The dose-response model used for calculating the probability of becoming infected in relation to the virus exposure is provided in 5.3.4, and all the parameters in QVEmod are listed in Table 3.

**5.3.1 State variables and scales.** QVEmod has two classes, the agents (individuals) and the environment. Both classes acquire virus over the course of a simulation. Individuals have 4 state variables: virus contamination on hands ($V_{hand}$), and the accumulated virus exposure via aerosols, droplets, and fomites ($E_{aerosols}$, $E_{droplets}$, $E_{fomites}$). The environment is composed of two air layers and one surface layer, all of which are divided into equally sized two-dimensional grid cells, the size of which is set to 0.25 m$^2$: a proximate of the space occupied by a single person. Each layer has a coordinate variable and a state variable to record the virus contamination in space ($V_{aerosols}$, $V_{droplets}$, $V_{fomites}$). The seven processes are evaluated each time step, which is configurable and set to the default value of 0.5 minutes. The default value of the time step is selected considering the rate of the processes in the model (e.g., for a given grid cell size, the time step should be small enough to capture the airflow between grid cells), and the model validation tests conducted with even smaller time step values show negligible differences in infection risk results.

**5.3.2 Input and initialisation.** QVEmod needs input for individuals' identifiers and movement scripts, both of which are generated by the activity scheduler and NOMAD model.

**Table 3. Parameters for the transmission model.**

| Attribute parameters | Value | Source |
|---|---|---|
| Emission rate ($\omega$) | Scaled to 1 unit per hour (Typical infectious individual, half breathing and half talking) | |
| Emission quantity by an average infectious individual ($\phi$) | $10^6$ RNA copies per hour (used for informing dose-response relationships) | [103] |
| Respiratory activity scaler ($\delta$) | 0.14 breathing<br>2.4 singing<br>1.86 talking<br>(relative to the baseline) | [56] |
| Individual Infectiousness scaler ($\sigma$) | 1 (A typical infectious individual)<br>0 (Susceptible individual) | |
| Proportion of viruses emitted in the form of aerosols ($p_{aerosols}$) | 0.978 (Breathing)<br>0.0652 (Singing)<br>0.171 (Talking) | [97] |
| Proportion of pathogen excreted to hands ($\eta$) | 0.15 | [27,104] |
| Transfer efficiency between hands and surfaces ($\theta$) | 0.25 per touch | [105,106] |
| Ratio of finger pads size to the cell size ($\pi$) | 0.0196 | Calculated based on [107,108,109] (see details in Section E in S1 Text) |
| Surface touching frequency ($\gamma$) | Tables: 15 touches per hour | [110] |
| Fractional transfer rate from hands to facial membranes ($\varepsilon$) | 0.01 per hour | Calculated based on various references (see details in Section E in S1 Text) |
| Unit decay rate of viruses in aerosols ($\mu_{aerosols}$) | 1.5 per hour | Set based on [111,112,113,114] (see details in Section E in S1 Text) |
| Unit deposition rate of droplets ($\mu_{droplets}$) | 37.93 per hour | [115,116] |
| Diffusion coefficient ($D$) | 0.0016 m$^2$/sec | [117] |
| Unit decay rate of viruses on surfaces ($\mu_{surfaces}$) | Wood: 0.969 per hour | [118] |
| Inhalation rate ($\rho$) | 288 L per hour (breathing, talking)<br>432 L per hour (singing) | [119,120,121] |
| Volume of a cell ($L$) | 125 L | |
| Infectious dose ($D_{inf}$) | 1000 RNA copies | [68] |
| The proportion of virions reaching respiratory cells $c_{aerosols}$, $c_{droplets}$, $c_{fomites}$ | 10% (aerosols)<br>10% (droplets)<br>10% (fomites) | Set in this paper (analysis results presented in Table 1) |
| Air change rate ($ACH$) | Air in a room is replaced 3 times per hour | [61] |
| Face mask filter efficiency aerosols ($FE_{aerosols}$) | 40% | [62] |
| Face mask filter efficiency droplets ($FE_{droplets}$) | 94% | [62] |

The latter contains the whereabouts and actions of each individual at each simulated time step, hence containing the duration of stay for each individual. In addition, the individuals' infectiousness status is generated randomly. Under the default setting, only one infected individual enters a simulation with an infectiousness scaler set to unity. Super shedders can be included as well, through the generation of a higher infectiousness scaler. By default, an individual's emission rate is based on breathing and talking at equal proportions, but other respiratory activities can be incorporated as well by the respiratory activity scaler. The size of the indoor space (width and length), and the location, size, and material of objects in the environment are user-defined inputs. In addition, interventions such as wearing masks, cleaning surfaces, and 1.5-meter physical distancing can be included as input to the model, which incurs changes in the activity scheduler, environment variables or NOMAD model parameters, respectively.

All state variables are initialised at zero, both for the environment ($V_{aerosols}$, $V_{droplets}$, $V_{fomites}$) and for individuals' exposure to the virus through either of the three transmission routes ($E_{aerosols}$, $E_{droplets}$, $E_{fomites}$). The superscripts $i$ and $s$ will be used to identify infectious agents and

susceptible agents when differentiation between agent groups is required for some variables. Susceptible individuals are initialised with virus contamination of zero on their hands ($V^s_{hands}$), whereas the contamination on infectious individuals' hands ($V^i_{hands}$) is initialised as a proportion of their emission rate, which is detailed in the following sections. All parameters used in the QVE model and their reference sources are presented in Table 3.

**5.3.3 Processes and agent-based state calculations.** Here, we describe the details of each of the seven processes (Fig 11). All these seven processes are continuous events and calculated for each time step ($\Delta t$) throughout the simulation.

*Infectious individuals emit virus into the air.* Infectious individuals emit viral-laden particles by speaking, coughing, or sneezing. As a result of virus emission, it is assumed that a portion $\eta$ of the pathogen is excreted to infectious agents' hands, whereas the rest is emitted to the air. The total amount of virus emitted and the partition of aerosols and droplets emitted to the air varies by respiratory activity (section 3.1.2). In this model, we assumed that aerosols are buoyant aerosols (d < 10um) and droplets constitute the rest of the particles (d > 10um). Infectious individuals are assumed to emit viruses at a constant rate. The unit of viral quantities used in this model follows from the typical emission of one typical infectious individual per time unit (default: hour). The virus emission calculation is triggered only for the cell *(x,y)* in which the infectious agent is at time *t*, otherwise, it is 0. The virus emission rate that infectious agent *i* shed into the air per time distributed over aerosols ($r^i_{emission\text{-}aerosols}$) and droplets ($r^i_{emission\text{-}droplets}$) are:

$$r^i_{emission-aerosols} = \omega(1-\eta)\delta\sigma p_{aerosols}(1 - FE_{aerosols})\Delta t \tag{8}$$

$$r^i_{emission-droplets} = \omega(1-\eta)\delta\sigma p_{droplets}(1 - FE_{droplets})\Delta t. \tag{9}$$

Here, $\omega$ represents the rate at which a typical infectious individual emits virus under half time breathing and talking condition, and is scaled to 1 per hour. $\eta$ represents the proportion of pathogen secreted to hands, therefore (1-$\eta$) represents the proportion emitted to the air. $\delta$ represents the activity infectiousness scaler for scaling the heterogeneity in emission rates during different respiratory activities, which scales the emission rate relative to the emission rate under half breathing and half talking condition. The infectiousness scaler, $\sigma$, scales different infectiousness levels of individuals relative to a typical emitter. $p_i$ represent the proportion of viruses emitted in the form of aerosols and droplets, where the two proportions ($p_{aerosols}$, $p_{droplets}$) add up to 1. $FE_i$ represent the filter efficiency of face masks for droplets or aerosols.

*Viral-laden droplets fall onto surfaces.* Viral-laden droplets can fall onto surfaces through sedimentation. The resulting contaminated surfaces are called fomites. We assume surfaces can acquire viruses from droplets. On surfaces, viruses are assumed to be stationary and evenly distributed within the grid cells. The rate of viruses transferring from droplets onto fomites ($r_{sedimentation}$) for cell *(x,y)* at time *t* is modelled as

$$r_{sedimentation}(x, y, t) = V_{droplets}(x, y, t)\mu_{droplets}\Delta t, \tag{10}$$

where $\mu_{droplets}$ represents the unit deposition rate of viral-laden droplets.

*Virus decay in the air and on surfaces.* SARS-CoV-2 viruses are assumed to decay exponentially in the environment, the rates of which vary in aerosols and on different surface materials. Viruses-laden aerosols lose infectivity at a constant rate while floating in the air, and air change rate (ACH) indoors has an increasing impact on their decay. Conversely, viruses-laden droplets are assumed to fall onto surfaces rapidly (Eq 10), so the decay in the droplet layer is assumed to be negligible. On fomites, viruses decay at a constant rate which depends on the fomite's material. The aerosols decay ($r_{decay\text{-}aerosols}$) and fomites decay ($r_{decay\text{-}fomites}$) equations

for cell *(x,y)* at time *t* is identified below where $\mu_{aerosols}$ and $\mu_{fomites}$ represent the unit decay rate of viruses in aerosols and on fomites respectively:

$$r_{decay-aerosols}(x,y,t) = V_{aerosols}(x,y,t)(1 - e^{-\mu_{aerosols}\Delta t - ACH\Delta t}) \tag{11}$$

$$r_{decay-fomites}(x,y,t) = V_{fomites}(x,y,t)(1 - e^{-\mu_{fomites}\Delta t}). \tag{12}$$

*Virus-laden aerosols and droplets diffuse in the air.* To simulate the diffusion of virus-laden particles in the air, we solve two-dimensional diffusion equations for the number of virions in aerosols and droplets. We assume that all particles are well-mixed in the volume of the grid cell, after which the aerosols start to diffuse in *x,y* directions (see Eqs 13–14). *Δx* and *Δy* represent the length unit of the cell (both 0.5m in the default). Here, *D* is the diffusion coefficient, indicating the unit diffusion rate per time (m$^2$/sec). The diffusion-induced rate of change in cell *(x,y)* at time *t* in aerosols ($r_{diffusion-a}(x,y,t)$) and droplets ($r_{diffusion-d}(x,y,t)$) are calculated with the equations below (for convenience in the representation, "aerosols" and "droplets" are abbreviated here as "*a*" and "*d*" respectively):

$$r_{diffusion-a}(x,y,t)$$
$$= D\frac{(V_a(x-\Delta x,y,t) + V_a(x+\Delta x,y,t) + V_a(x,y-\Delta y,t) + V_a(x,y+\Delta y,t) - 4V_a(x,y,t))\Delta t}{\Delta x \Delta y} \tag{13}$$

$$r_{diffusion-d}(x,y,t)$$
$$= D\frac{(V_d(x-\Delta x,y,t) + V_d(x+\Delta x,y,t) + V_d(x,y-\Delta y,t) + V_d(x,y+\Delta y,t) - 4V_d(x,y,t))\Delta t}{\Delta x \Delta y}. \tag{14}$$

*Susceptible individuals inhale air with viral-laden droplets and aerosols.* Susceptible individuals get exposed to the virus from aerosols and droplets by inhaling a portion of airborne viruses accumulated in the air ($V_{aerosols}(x,y,t)$ and $V_{droplets}(x,y,t)$) in the cell *(x,y)* they are in at time *t*. For each susceptible agent *s*, we calculate the inhaled amount of viruses per time step via aerosols and droplets by $r^s_{inhalation-aerosols}(t)$ and $r^s_{inhalation-droplets}(t)$, respectively. Then, again for each susceptible agent *s*, the accumulated virus exposure via aerosols and droplets, $E^s_{aerosols}(T)$ and $E^s_{droplets}(T)$ are calculated by the summation of the inhaled amount of viruses up to time *T*. The inhalation of virus in the forms of aerosols and droplets is the ratio of human tidal volume per time step over the cell volume (*L*), where *ρ* represents the unit inhalation rate, which depends on the respiratory activities of an individual. $FE_i$ represents the filter efficiency of face masks against aerosols or droplets.

$$r^s_{inhalation-aerosols}(t) = V_{aerosols}(x,y,t)\frac{\rho}{L}(1 - FE_{aerosols})\Delta t \tag{15}$$

$$r^s_{inhalation-droplets}(t) = V_{droplets}(x,y,t)\frac{\rho}{L}\left(1 - FE_{droplets}\right)\Delta t \tag{16}$$

$$E^s_{aerosols}(T) = \sum_{t=0}^{T} r^s_{inhalation-aerosols}(t) \tag{17}$$

$$E^s_{droplets}(T) = \sum_{t=0}^{T} r^s_{inhalation-droplets}(t) \tag{18}$$

*Infectious individuals contaminate surfaces.* Infectious people can contaminate surfaces by interacting with them. It is assumed that virus on infectious people's hands, $V^i_{hand}$, can be

transferred to surfaces. Surfaces, such as tables and chairs in cell *(x,y)* are assumed to be touched by proximate individuals at a constant rate if there is a surface area within the reachable distance (0.5 m) of the infectious agent, *i*. For a grid cell *(x,y)* containing surface elements, the touching frequency ($\gamma$), transfer efficiency ($\theta$), and the ratio of finger pads surface relative to the reachable surface area ($\pi$) determines the surface contamination rate in a time step, $r^i_{contamination}(x,y,t)$. $V^i_{hand}$ is initialised at *t* = 0 as a proportion of emission rate, where $\eta$ represents the proportion of pathogen excreted to hands. It is assumed that the decrease rate of the virus on the infectious agent's hands (due to decay or transfer) is similar to its replenishment rate, then the change in the virus amount on the infectious agent's hands is negligible. Hence, $V^i_{hand}$ is assumed to be constant throughout the event:

$$r^i_{contamination}(x,y,t) = V^i_{hand}(t)\gamma\theta\pi\Delta t \tag{19}$$

$$V^i_{hand}(t) = V^i_{hand}(0) = \omega\eta. \tag{20}$$

*Susceptible individuals touch virus on the surfaces.* Susceptible individuals' exposure to the virus from fomites is the amount of virus on fomites being picked up by their hands and sent to their facial membranes. It is assumed that, first, the virus transfer from surfaces to hands occurs when susceptible people touch the contaminated surface at cell *(x,y)*, and the virus accumulates in each susceptible agents' hand, $V^s_{hand}$. Then, again for each susceptible agent *s*, the individual exposure from fomites route up to time *T*, $E^s_{fomites}(T)$, is calculated as a proportion of viruses on hands that are assumed to be transferred from hands to facial membranes, $\varepsilon$. Similar to the surface contamination process, the touching frequency ($\gamma$), transfer efficiency ($\theta$), and the ratio of finger pads relative to the reachable surface area ($\pi$) are used to calculate the virus pick up rate.

$$r^s_{pick-up}(x,y,t) = V_{fomites}(x,y,t)\gamma\theta\pi\Delta t \tag{21}$$

$$r^s_{pick-up}(t) = \sum_{x,y} r^s_{pick-up}(x,y,t) \tag{22}$$

$$V^s_{hand}(t+\Delta t) = V^s_{hand}(t) + r^s_{pick-up}(t) \tag{23}$$

$$E^s_{fomites}(T) = \sum_{t=0}^{T} V^s_{hand}(t)\varepsilon\Delta t \tag{24}$$

**5.3.4 Environmental state calculations.** As a result of the processes explained above, the state variables in the environment $V_{aerosols}$, $V_{droplets}$, $V_{fomites}$ are calculated and updated for each grid cell *(x,y)* in each *Δt*.

In each time step *Δt*, $V_{aerosols}$ is decreased by the inhaled amount by the susceptible agents in grid cell *(x,y)*, updated by the diffused amount of particles, decreased by the decay of viruses and increased by the virus emission if there exists an infectious agent in cell *(x,y)* at time *t*:

$$V_{aerosols}(x,y,t+\Delta t)$$
$$= V_{aerosols}(x,y,t) - \sum_s r^s_{inhalation-aerosols}(t) + r_{diffusion-aerosols}(x,y,t) - r_{decay-aerosols}(x,y,t)$$
$$+ r^i_{emission-aerosols}. \tag{25}$$

Similarly, $V_{droplets}$ is decreased by the inhaled amount by the susceptible agents in grid cell *(x,y)*, updated by the diffused amount of particles, decreased by the sedimentation of viruses from air layer to surface layer, and increased by the virus emission if there exists infectious

agent in cell *(x,y)* at time t:

$$V_{droplets}(x, y, t + \Delta t)$$
$$= V_{droplets}(x, y, t) - \sum_s r^s_{inhalation-droplets}(t) + r_{diffusion-droplets}(x, y, t) - r_{sedimentation}(x, y, t)$$
$$+ r^i_{emission-droplets}. \tag{26}$$

In the surface layer, $V_{fomites}$ is decreased by the picked-up amount by the susceptible agents within the reachable distance to grid cell *(x,y)*, increased by the sedimentation of viruses from air layer to surface layer, decreased by the decay of viruses on the surfaces and increased by the virus contamination if there exists an infectious agent within the reachable distance to grid cell *(x,y)* at time *t*:

$$V_{fomites}(x, y, t + \Delta t)$$
$$= V_{fomites}(x, y, t) - \sum_s r^s_{pick-up}(x, y, t) + r_{sedimentation}(x, y, t) - r_{decay-fomites}(x, y, t)$$
$$+ r^i_{contamination}(x, y, t). \tag{27}$$

**5.3.5 Estimating infection risks.** QVEmod calculates each individual's exposure via three routes $E^s_{aerosols}$, $E^s_{droplets}$, and $E^s_{fomites}$. Recall that the magnitude of $E^s$ variables are scaled since the unit emission rate $\omega$ is initially scaled to 1 for computational purposes. Therefore, the number of viral particles someone is exposed to is rescaled as a product of $E^s$ variables and $\phi$, the emission rate by an average infectious individual (see Table 3).

The relationship between the number of viral particles someone is exposed to, and the risk of acquiring infection is likely to differ between transmission routes, because of different deposition locations (faces, lower and upper respiratory tract) and the viability of the virus, among others [122,123]. Accordingly, we modelled the relationship between the three exposure routes and the infection risk using an exponential dose-response relationship [124] as below:

$$P^s = 1 - e^{-\left(\frac{\phi E^s_{aerosols}(T)}{k_{aerosols}} + \frac{\phi E^s_{droplets}(T)}{k_{droplets}} + \frac{\phi E^s_{fomites}(T)}{k_{fomites}}\right)}. \tag{28}$$

where $P^s$ represents the susceptible individual's probability of getting infected, $E^s_{aerosols}(T)$, $E^s_{droplets}(T)$, $E^s_{fomites}(T)$ the individual's scaled accumulated exposure via the three transmission routes, $\phi$ the emission rate by an average infectious individual, and $k_{aerosols}$, $k_{droplets}$, $k_{fomites}$ the route specific exposure parameter, which corresponds to an exposure level resulting in 63% chance of getting infected via an individual route. The $k_{route}$ depends on the infectious dose $D_{inf}$, for which we consider recent estimates of the founding virus population size required to cause infection in a recipient host [68] and the proportion of viral particles someone is exposed to that reach the respiratory tract cells ($c_{route}$) and thus contribute to the founding population:

$$k_{route} = \frac{D_{inf}}{c_{route}}. \tag{29}$$

Here, $c_{route}$ is an unknown parameter and particularly hard to estimate. We therefore explore a range of different options in the Results section.

We then used the calculations for individual exposures to estimate the number of infected individuals that occurred during a specific event:

- Using each individual's cumulative exposure, the dose-response model provides an estimate for infection risk: $P^s$, the probability that the susceptible individual s acquired an infection during their stay.

- Then, for each susceptible individual in the simulation, a random number from the uniform distribution [0,1] is drawn, and this random number is compared to the individual's infection probability. If the individual's infection probability was larger than the number drawn, then it is assumed that an infection is realised.

- The total number of new infections that occurred during a specific scenario was estimated by the summation of infections realised.

- We repeated this 10,000 times to obtain a distribution of the number of infections that may have occurred.

- The mean of this distribution can be regarded as the event-specific reproduction number R: the average number of new infections that arose from one specific event with one infectious individual present.

The parameter values used in QVEmod are depicted in Table 3. These reflect the most recent insights about SARS-CoV-2 characteristics, and can be configured with respect to new information available. For a detailed description of the parameterization, the reader is referred to Section E in S1 Text.

## Supporting information

**S1 Text. Supporting Information.** Section A. Background information on indoor movement and transmission models. Section B. Experiment setting for static contacts. Section C. Description of SamenSlimOpen tool. Section D. Case study description. Section E. Parameter description in QVE-MOD. References for supporting information. **Fig A. Screenshots of the SamenSlimOpen tool.** A) introduction screen, B) scene selection screen, C) scene development screen, D) developed scenario. **Fig B. The case study restaurant layout.** The green rectangles and round brown circles signify the seats, the green arrows the entrances, the blue toilets the entrance to the toilets and the brown rectangles the tables. Fig C. The snapshot contamination map in the case study. Virus contamination in the environment in aerosols, droplets, and on fomites over time in minutes. Contamination is expressed as the virion quantity relative to an average infectious individual's hourly emission. **Fig D. The contamination maps in the case study for ventilation and face mask scenarios.** (A,B,C) are the scenarios where individuals do not wear face masks and ACH is 0 per hour in the restaurant in (A), 3 in (B), and 6 in (C). (D, E) are the scenarios where people wear face masks while moving and ACH is 3 per hour in the restaurant in (D) and 6 in (E). Within each scenario, the impact of intervention on viral spread is presented: (a, b, c) show virus concentration in the aerosols, droplets, and fomites, respectively. **Fig E. The analysis of relative contribution of transmission routes in the case study.** Each row shows a parameters setting for $c_{route}$ (A) $c_{route}$ is the same for all routes ($c_{aerosols}$:$c_{droplets}$:$c_{fomites}$ is 10%:10%:10%). (B) $c_{route}$ is smaller for fomites ($c_{aerosols}$:$c_{droplets}$:$c_{fomites}$ is 10%:10%:1%). (C) $c_{route}$ is smaller for fomites and droplets ($c_{aerosols}$:$c_{droplets}$:$c_{fomites}$ is 10%:1%:1%). (D) $c_{route}$ is smaller for fomites and aerosols ($c_{aerosols}$:$c_{droplets}$:$c_{fomites}$ is 1%:10%:1%). Each column shows an intervention scenario: (a) poor ventilation scenario, ACH = 0, (b) baseline scenario, ACH = 3, (c) scenario with recommended ventilation, ACH = 6, (d) baseline scenario with face masks worn while moving, (e) scenario with recommended ventilation and with face masks worn while moving. **Fig F. Sensitivity analysis of emission rate.** The distributions of the expected number of infected individuals in the case study with different emission quantities $\phi$. (A) to (E) show the results for changing $\phi$ values from 10^5 to 10^7. This may reflect the heterogeneity in viral load of the index patients. The black solid lines indicate the mean value of the infected

number in the baseline scenario and the dashed lines show the mean value corresponding to each respective scenario. **Fig G. Sensitivity analysis of proportions of aerosols.** The distributions of the expected number of infected individuals in the case study with different proportions of virus emitted in the form of aerosols. In the baseline scenario, $p_{aerosols}$ is 22.91%. (A) to (E) shows the results for from 50% lower to 50% higher (namely 11.45%, 17.18%, 22.91%, 28.63%, 34.37%) representing the heterogeneity due to respiratory activities or individual variation. The black solid lines indicate the mean value of the infected number in the baseline scenario and the dashed lines show the mean value corresponding to each respective scenario. **Fig H. Sensitivity analysis of virus decay rate on surfaces.** The distributions of the expected number of infected individuals in the case study with different virus decay rates on surfaces. In the baseline scenario $\mu_{surfaces}$ for wood is 0.969 per hour. (A) to (E) shows the results for changing $\mu_{surfaces}$ from 90% lower to 90% higher (namely 0.0969, 0.4845, 0.969, 1.4535, 1.8411 per hour) representing the heterogeneity due to different surface materials. The black solid lines indicate the mean value of the infected number in the baseline scenario and the dashed lines show the mean value corresponding to each respective scenario. **Fig I. Sensitivity analysis of virus transfer rate between hand and surface.** The distributions of the expected number of infected individuals in the case study with different diffusion rates $D$ and virus decay rates in aerosols $\mu_{aerosols}$. Each row shows a parameter setting for diffusion: (A) Diffusion rate is 0.000278 m$^2$/s, 6 times smaller than the baseline scenario. (B) Diffusion rate is at the baseline scenario 0.0016m$^2$/s. (C) Diffusion rate is 0.01 m$^2$/s as an upper bound from literature (Kudryashova et al. 2021), 6 times larger than the baseline scenario. Each column shows a parameter setting for virus decay rate in aerosols $\mu_{aerosols}$. (a) Decay rate is 0.755/hour, 50% lower than the baseline scenario (b) Decay rate is 1.51/hour as the baseline scenario. (c) Decay rate is 2.27/hour, 50% higher than the baseline scenario. The black solid lines indicate the mean value of the infected number in the baseline scenario and the dashed lines show the mean value corresponding to each respective scenario. **Fig J. Sensitivity analysis of fractional virus transfer rate from hand to facial membranes.** The distributions of the expected number of infected individuals in the case study with different diffusion rates $D$ and deposition rates $\mu_{droplets}$. Each row shows a parameter setting for diffusion: (A) Diffusion rate is 0.000278 m$^2$/s, 6 times smaller than the baseline scenario. (B) Diffusion rate is at the baseline scenario 0.0016m$^2$/s. (C) Diffusion rate is 0.01 m$^2$/s as an upper bound from literature (Kudryashova et al. 2021), 6 times larger than the baseline scenario. Each column shows a parameter setting for deposition: (a) Deposition rate is 18.97/hour, 50% lower than the baseline scenario (b) Deposition rate is 37.93/hour as baseline scenario. (c) Deposition rate is 56.90/hour, 50% higher than the baseline scenario. The black solid lines indicate the mean value of the infected number in the baseline scenario and the dashed lines show the mean value corresponding to each respective scenario. **Fig K. Sensitivity analysis of diffusion rate and deposition rate.** The distributions of the expected number of infected individuals in the case study with different virus transfer rates between hand and surface ($\theta\pi\gamma$). The baseline transfer rate between hand and surface is 0.0735 (0.0196*0.25*15) per hour. (A) to (E) shows the results for changing transfer rates from 75% lower to 75% higher (namely 0.0184, 0.0368, 0.0735, 0.1103, 0.1286 per hour) representing the heterogeneity of touching surface behaviour. The black solid lines indicate the mean value of the infected number in the baseline scenario and the dashed lines show the mean value corresponding to each respective scenario. **Fig L. Sensitivity analysis of diffusion rate and virus decay rate in aerosols.** The distributions of the expected number of infected individuals in the case study with different fractional virus transfer rates from hand to facial membranes $\varepsilon$. The baseline transfer rate from hand to facial membranes is 1%. (A) to (E) shows the results for changing $\varepsilon$ from 0.1%, 0.5%, 1%, 5% and 10% representing the

heterogeneity of touch face behaviour. The black solid lines indicate the mean value of the infected number in the baseline scenario and the dashed lines show the mean value corresponding to each respective scenario.
(PDF)

## Acknowledgments

We thank Bas Dado, Wim van der Poel, Rineke de Jong, Marion Koopmans, Sander Herfst, Alexander Verbraeck, Yilin Huang, and Els van Daalen for their discussions and support in both research and acquisition. Moreover, we thank the testers of the SSO app for their enthusiasm and critical notes.

## Author Contributions

**Conceptualization:** You Chang, Linda van Veen, Dorine Duives, Quirine A. ten Bosch.

**Data curation:** Büsra Atamer Balkan, You Chang, Yangfan Liu.

**Funding acquisition:** Reina S. Sikkema, Linda van Veen, Dorine Duives, Quirine A. ten Bosch.

**Methodology:** Büsra Atamer Balkan, You Chang, Martijn Sparnaaij, Berend Wouda, Doris Boschma, Yangfan Liu, Mart C. M. de Jong, Colin Teberg, Kevin Schachtschneider, Reina S. Sikkema, Dorine Duives, Quirine A. ten Bosch.

**Software:** Büsra Atamer Balkan, Martijn Sparnaaij, Berend Wouda, Doris Boschma, Colin Teberg, Kevin Schachtschneider.

**Supervision:** Winnie Daamen, Linda van Veen, Dorine Duives, Quirine A. ten Bosch.

**Visualization:** You Chang.

**Writing – original draft:** Büsra Atamer Balkan, You Chang, Dorine Duives, Quirine A. ten Bosch.

**Writing – review & editing:** Martijn Sparnaaij, Yufei Yuan, Winnie Daamen, Mart C. M. de Jong, Reina S. Sikkema, Linda van Veen.

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
