## [Decision Letter · Decision Letter 0]

27 Oct 2023

Dear Dr. ten Bosch,

Thank you very much for submitting your manuscript "The multi-dimensional challenges of controlling respiratory virus transmission in indoor spaces -  Insights from the linkage of a microscopic pedestrian simulation and SARS-CoV-2 transmission model" for consideration at PLOS Computational Biology.

As with all papers reviewed by the journal, your manuscript was reviewed by members of the editorial board and by several independent reviewers. In light of the reviews (below this email), we would like to invite the resubmission of a significantly-revised version that takes into account the reviewers' comments.

We cannot make any decision about publication until we have seen the revised manuscript and your response to the reviewers' comments. Your revised manuscript is also likely to be sent to reviewers for further evaluation.

Sincerely,

Benjamin Althouse

Academic Editor

PLOS Computational Biology

Virginia Pitzer

Section Editor

PLOS Computational Biology

Reviewer's Responses to Questions

**Comments to the Authors:**

Reviewer #1: The paper describes a thorough mechanisitc modelling study of COVID transmission in a restaurant setting. The work uses carefully considered and integrated sub-models to represent activity, movement, viral exposure and risk. The authors should be congratulated on an important and comprehensive study. I believe it adds an important element to the body of work on viral transmission and mitigation and I support its publication.

There are a number of issues that should be addressed before publication and these are listed below.

Major points:

* The model includes three routes of transmission, including fomite. However, as the authors say, the baseline scenario does not favour this route as there are no shared surfaces in the scenario. This is made clear in the interpretation. However, for the main scenario the modelling of surface touch is a little hard to assess. If I have understood correctly, the tables and chairs around them are the surfaces that can be contaminated and shared, effectively acting as a transmission route. That seems reasonable but doesn't seem to cover all of the potential for cross contamination. Many studies have focused on high touch surfaces such as door handles that have the potential to pass contamination to multiple users of the space. This might be appropriate in this scenario for the entrance/exit as well as for the bathrooms (and the other surfaces therein). Also, are the coat rack and pay register included as touchable surfaces? It would be good to clarify this and to discuss it in the limitations section.

* Although there is an exploration of the uncertainty in the relationship between viral exposure and risk of infection that includes different levels of susceptibility, the assumption of average infectivity is not explored particularly. Studies suggest that the viral load varies over many orders of magnitude. This is likely to have a strong effect on risk and is best thought of as a continuous spectrum rather than normal and superspreader. It would be good to acknowledge this as a central source of uncertainty to be borne in mind when interpreting these results.

* p24, eqn(23) - why does the amount of virus on the hand not diminish through transfer to mucous membranes? On a related note, the epsilon parameter in Table 3 is given with units of per hour but has no such units in the supplementary information. This value does seem very low. The SI doesn't fully explain how it is derived. Could this be included for transparency.

Minor points:

p3, l114 - Wells-Riley (and elsewhere)

p3, l146 - '... across the range of possible...'?

p9, l357 - '... for an extended period of time...'

p12, fig 8 - the y-axis should have correct labels (Infection risk?) and the existing ones moved perhaps horizontally away from the axes to avoid confusion.

p14, l498 - '...less efficiently...'

p16, l606 - 'Nomad' and 'NOMAD' used interchangably in numerous places

p16, l617 - sentence doesn't make sense - too many words or words missing?

p26, Table 3 - What does 'Proportion of pathogen exerced to hands' mean? Is that a typo?

p27, Table 3 - The breathing rates look low for adults. I would expect around 600 L hr-1. There are tabulated values widely used for environmental assessment rather than calculating from tidal breathing and rate of breathing.

Reviewer #2: I commend the authors for a detailed and well-crafted manuscript. The PeDVis model developed by the authors is an effective way to limit the transmission of respiratory viruses in future pandemics. A crucial aspect of pandemic management plans is the implementation of interventions aimed at reducing transmission within indoor spaces. However, I have a few suggestions for minor corrections that could be made in the manuscript.

Lines 70-71 state that the duration, closeness, and number of contacts the infectious individual has while visiting the indoor space drive transmission. This statement is accurate, but it would benefit from being supported by a reference.

Authors should cite multiple studies when using the phrase "These studies show" in line 73 (lines 71-75).

In line 228, I suggest that "breath" should be rewritten as "breathe" to make the sentence clearer and grammatically correct.

**Have the authors made all data and (if applicable) computational code underlying the findings in their manuscript fully available?**

Reviewer #1: Yes

Reviewer #2: None

PLOS authors have the option to publish the peer review history of their article (what does this mean?). If published, this will include your full peer review and any attached files.

Reviewer #1: No

Reviewer #2: **Yes: **Lawrence Annison
---

## [Decision Letter · Decision Letter 1]

19 Feb 2024

Dear Dr. ten Bosch,

Thank you very much for submitting your manuscript "The multi-dimensional challenges of controlling respiratory virus transmission in indoor spaces: Insights from the linkage of a microscopic pedestrian simulation and SARS-CoV-2 transmission model" for consideration at PLOS Computational Biology. As with all papers reviewed by the journal, your manuscript was reviewed by members of the editorial board and by several independent reviewers. The reviewers appreciated the attention to an important topic. Based on the reviews, we are likely to accept this manuscript for publication, providing that you modify the manuscript according to the review recommendations.

Reviewer 1 has a few small suggestions for further edits. Once these have been addressed, we should be able to accept the manuscript without further review.

Sincerely,

Benjamin Althouse

Academic Editor

PLOS Computational Biology

Virginia Pitzer

Section Editor

PLOS Computational Biology

Reviewer's Responses to Questions

**Comments to the Authors:**

Reviewer #1: Thank you to thte authors for addressing the comments so carefully and thoroughly. I was happy with all of the comments with the exception of the response below where I had one further suggestion.

R1 - comment 1 response. The addition to the text is helpful but is slightly circular - it seems to say COVID-19 mainly is spread via other routes so we haven't dealt with shared surfaces properly. I'm oversimplifying and I think the final sentence is helpful pointing to these other surfaces. I would ask though that the second and third sentences are revised. Other studies have shown that high touch surfaces can be important for transmission. For example, Miller et al. (2022), have shown with modelling of a shared space that touch of only briefly contaminated surfaces can be important when taken over large numbers of people. Experimental measurement of the variability of surface contamination in real environments is difficult and I believe the current wording is too definitive. I would suggest the following ammendment or similar:

"However, for COVID-19, evidence shows that the virus mainly spreads through

respiration (Greenhalgh et al., 2021; Miller et al., 2021; Zhang et al., 2020), and transmission through

surfaces *may be* limited (Mondelli et al. 2020; Meyerowitz, Richterman, Gandhi, et al. 2021; Lewis et al.,

2021; Zhang et al., 2021; Cheng et al., 2022). Experimental studies in cats have shown that SARSCoV-

2 can be transmitted through the environment (Gerhards et al., 2023) but this transmission is

primarily associated with the accumulation of the virus in the environment over prolonged time in

a shared space rather than being linked to high-touch surfaces, although Miller et al. (2022) show that high-touch surfaces can be important in crowded environments."

Miller, Daniel, et al. "Modeling the factors that influence exposure to SARS‐CoV‐2 on a subway train carriage." Indoor Air 32.2 (2022): e12976.

In addition, on scanning the paper I noticed that two of the mechanisms in Fig 11 had the same number: 6. Inhalation and 6. Contaminate Surfaces.

Reviewer #2: I would like to commend the authors for their diligent efforts in significantly improving the manuscript.

**Have the authors made all data and (if applicable) computational code underlying the findings in their manuscript fully available?**

Reviewer #1: Yes

Reviewer #2: Yes

PLOS authors have the option to publish the peer review history of their article (what does this mean?). If published, this will include your full peer review and any attached files.

Reviewer #1: No

Reviewer #2: **Yes: **Lawrence Annison

Figure Files:

Data Requirements:

Reproducibility:

References:

---

## [Editor Report · Decision Letter 2]

29 Feb 2024

Dear Dr. ten Bosch,

We are pleased to inform you that your manuscript 'The multi-dimensional challenges of controlling respiratory virus transmission in indoor spaces: Insights from the linkage of a microscopic pedestrian simulation and SARS-CoV-2 transmission model' has been provisionally accepted for publication in PLOS Computational Biology.

Best regards,

Benjamin Althouse

Academic Editor

PLOS Computational Biology

Virginia Pitzer

Section Editor

PLOS Computational Biology

---

## [Editor Report · Acceptance letter]

25 Mar 2024

PCOMPBIOL-D-23-01128R2 

The multi-dimensional challenges of controlling respiratory virus transmission in indoor spaces: Insights from the linkage of a microscopic pedestrian simulation and SARS-CoV-2 transmission model

Dear Dr ten Bosch,

I am pleased to inform you that your manuscript has been formally accepted for publication in PLOS Computational Biology. Your manuscript is now with our production department and you will be notified of the publication date in due course.

With kind regards,

Bernadett Koltai
